**Subject Category:**
Biology (whole organism)

biophysics/biomaterials

biomineralization, barnacle, plate, growth, structure, morphology

**Author for correspondence:**
Antonio G. Checa
e-mail: acheca@ugr.es

# Articulation and growth of skeletal elements in balanid barnacles (Balanidae, Balanomorpha, Cirripedia)

Antonio G. Checa[1,2], Carmen Salas[3], Alejandro B. Rodríguez-Navarro[4], Christian Grenier[1] and Nelson A. Lagos[5]

[1]Departamento de Estratigrafía y Paleontología, Universidad de Granada, 18071 Granada, Spain
[2]Instituto Andaluz de Ciencias de la Tierra, CSIC-Universidad de Granada, 18100 Armilla, Spain
[3]Departamento de Biología Animal, Facultad de Ciencias, Universidad de Málaga, 29071 Málaga, Spain
[4]Departamento de Mineralogía y Petrología, Universidad de Granada, 18071 Granada, Spain
[5]Centro de Investigación e Innovación para el Cambio Climático, Universidad Santo Tomás, Santiago, Chile

AGC, 0000-0001-7873-7545

The morphology and ultrastructure of the shells of two balanid species have been examined, paying special attention to the three types of boundaries between plates: (i) radii-parietes, (ii) alae-sheaths, and (iii) parietes-basal plate. At the carinal surfaces of the radii and at the rostral surfaces of the alae, there are series of crenulations with dendritic edges. The crenulations of the radius margins interlock with less prominent features of the opposing paries margins, whereas the surfaces of the longitudinal abutments opposing the ala margins are particularly smooth. The primary septa of the parietes also develop dendritic edges, which abut the internal surfaces of the primary tubes of the base plates. In all cases, there are chitino-proteinaceous organic membranes between the abutting structures. Our observations indicate that the very edges of the crenulations and the primary septa are permanently in contact with the organic membranes. We conclude that, when a new growth increment is going to be produced, the edges of both the crenulations and the primary septa pull the viscoelastic organic membranes locally, with the consequent formation of viscous fingers. For the abutting edges to grow, calcium carbonate must diffuse across the organic membranes, but it is not clear how growth of the organic membranes themselves is accomplished, in the absence of any cellular tissue.

# 1. Introduction

Barnacles (Infraclass Cirripedia) are crustaceans mostly characterized by an adult sessile life mode by attachment to hard substrata, including other shelled organisms. Non-thoracican barnacles have different, highly diverse life habits, including borers and parasites. Their growth mode is markedly different from the typical arthropod growth pattern, because they secrete external calcareous plates with incremental growth. Goose barnacles (former order Pedunculata) attach to the substrate by means of a stalk, which is absent in acorn barnacles (order Sessilia). The suborder Balanomorpha is the main constituent of the Sessilia, and its elements are characterized by a crown of calcareous plates (one, four, six or eight) arranged in a cone-like fashion (wall plates). In some groups, the soft parts attach directly to the substrate by means of an organic adhesive, whereas others attach by means of a calcareous basal plate (e.g. [1]). The continuously growing wall plates form a kind of shell enclosing the body; ecdysis is limited to the cuticle covering the soft body tissues inside the shell and at the attachment band [2–4].

The modifications made by sessile acorn barnacles to their crustacean structures for a predominantly sessile mode of life allow them both strong substrate adhesion and continuous radial growth [5]. These traits are partly responsible for their success in littoral environments. When measuring success by the criteria of either the density of individuals or geographical range, several species of barnacles must be ranked among the most successful colonizers of the marine littoral zone, particularly the so-called mid-littoral or 'barnacle' zone in the intertidal rocky zonation [6]. For similar reasons, sessile barnacles have become one of the most successful animals in biofouling, i.e. the accretion of marine organisms over man-made structures [7,8]. There is also increasing evidence that climate change is affecting species distributions, population dynamics and biology [9], particularly concerning the sessile organisms of the intertidal zone, such as barnacles. Several studies have assessed the importance of pH and temperature changes in several balanids [10–12].

According to the growth mode of the set of plates, growth in balanomorphs can be differentiated into two types. Monometric growth [13] proceeds with growth of the triangular wall plates at their basal (dorsal) margins, which also enlarge with time, such that the basal periphery increases at the same time that the form elevates above the substrate. In this type of growth, the ventrally placed aperture cannot expand and is enlarged by erosion of the apical margin [14]. The other growth pattern is diametric growth (initially defined by Darwin [15]), in which there is an additional increase in the circumference of the wall both apically and basally. This is achieved because the plates develop radii, which grow laterally in the direction towards the carina.

General descriptions of the morphological elements of the balanomorphs can be found in references [1,15–18]. The members of the Balanidae are characterized by diametric growth and may have four or six wall plates [19]. There is an anterior rostral and a posterior carinal plate. In six-plated balanids, the four remaining wall plates have been traditionally termed carinolateral (two) and lateral (two) plates, but due to lack of homology with similarly termed plates of scalpellomorphs, they have been renamed as carinomarginal and rostromarginal, respectively [18]. We will follow the latter terminology. Plates consist of two main basic elements: parietes and radii (figure 1a,b; electronic supplementary material, figures S1a,b, S2a,b and S3). The number and types of calcified elements are completed with two pairs of opercular plates: two anterior scuta and two posterior terga (figure 1a,b; electronic supplementary material, figure S3). The parietes are apex-up triangular areas, which grow at their bases, whereas the radii are apex-down triangular structures continuous with the parietes, which display lateral growth. In six-plated forms (with which this study is concerned), the rostral plate is the only one with two radii. The rostromarginal and carinomarginal plates have only one radius each at their carinal margins and no radius is present in the carinal plate (figure 1a–c; electronic supplementary material, figures S1a,b, S2a,b and video S1). Internally, each plate also develops a sheath which extends laterally into an ala (figure 1; electronic supplementary material, figure S2b). Contrary to the radii, the sheath of the carinal plate has two alae; the sheaths of the carinomarginals and rostromarginals have only one ala and the rostral sheath has none (figure 1b; electronic supplementary material, figure S2b and video S1). Whereas the sheaths grow towards the dorsum, the alae are characterized by lateral growth, towards the rostrum (figure 1b), which compensates for the lateral, carinal-ward growth of the radii. In the Balanidae, depending on the subfamily, the growth margins of the alae extend towards the internal surface of the neighbouring plate until making contact with its sheath margin or with a longitudinal ledge (longitudinal abutment) immediately posterior to the sheath margin [19]. Balanids have a calcareous basal plate, which

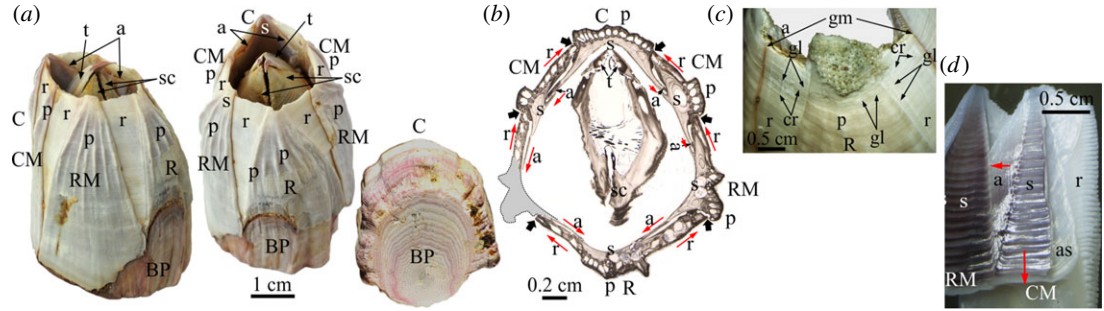

**Figure 1.** Main elements of the balanid shells, and terminology applied to them. (*a*) Right anterior, anterior upper (ventral) and lower (dorsal) views of a specimen of *Austromegabalanus psittacus*. (*b*) Transversal thin section of a methacrylate-embedded specimen of *A. psittacus*, close to the ventral side; the paries of the right rostromarginal plate is lacking (grey area). The red arrows indicate the growth directions of the radii and alae. The thick arrows indicate the outer boundaries between wall plates. (*c*) Partial ventrolateral view of a specimen of *A. psittacus*. (*d*) Interior of a specimen of *Perforatus perforatus*. The red arrows indicate the growth directions of the sheath and ala. a, ala; as, articular surface for the adjacent ala; BP, basal plate; C, carina; CM, carinomarginal plate; cr, crenulation; gl, growth line; gm, growth margin; p, paries; R, rostrum; r, radius; RM, rostromarginal plate; s, sheath; sc, scutum; t, tergum.

attaches the animal directly to the substrate [15,20] (figure 1*a*; electronic supplementary material, figures S1, S2*c* and video S1).

When trying to understand how this suite of plates grows, it is particularly important to study their growth margins. The elements found at the lateral and basal margins of the plates are named according to the nomenclature provided in [16,17,21]. Relatively few authors have dealt with this aspect. In some barnacles, the carinal margins of the plates develop a system of teeth and sockets which interlock tightly with a complementary system of much blunter teeth and shallower sockets of the rostral margin of the neighbouring plate. This system was described in the Balanidae *Balanus balanus* [5] and *Amphibalanus reticulatus* [22], and in the Tetraclitidae *Tesseropora rossea* [13]. Nevertheless, the same margins were found to be smooth in both the Archaeobalanidae *Semibalanus balanoides* [5] and the Balanidae *Membranobalanus declivis* [21]. Pitombo [21] described and figured a variety of 'septa' with and without denticles on the sutural edges of the radii of several other Balanidae, although he did not describe the opposing parietal growth margins. A comparable system of teeth and sockets was found at the contacts of the alae and the sheaths or the longitudinal abutments in *T. rossea* [13], *S. balanoides* [5] and *A. reticulatus* [22], but not in *B. balanus* [5].

The existence of an organic membrane (named as cuticular) was described between the junctions radius–paries in *B. balanus* [5]. Similar membranes were found at the radius–paries and ala–sheath junctions in *A. reticulatus* [22].

The basal margins of the wall plates also interlock with the margin of the basal plate [15,16]. Here, the basal denticulate ends of the longitudinal septa, the primary septa, which separate the longitudinal canals of the wall plates, penetrate the openings of the primary canals of the basal plate.

Other than these studies, the exact details of the interlocking systems between plates are not known. How exactly do structures at the plate boundaries interlock? How is the living tissue distributed at those plate-abutting boundaries? How does such a complex teeth-and-socket system develop? Are all boundaries morphologically equivalent? In summary, there is hitherto a lack of information about the features and processes taking place at a micrometre level, which is essential to understand how incremental growth of the balanoid skeleton proceeds. To answer these important questions, we have carried out a detailed study of the morphology, distribution and composition of the hard and soft parts constituting the growth margins of the shell and basal plates of two balanid species, using complementary analytical techniques such as electron and optical microscopy, micro-CT and infrared spectroscopy. The information obtained by comparing the different types of growth boundaries of plates provides important insight on the morphogenesis of the structures observed and on the precise growth mechanisms taking place at these boundaries. Accordingly, knowing the exact mode of cirriped plate growth is of interest in morphology and biomineralization. But it is also important in biofouling studies, which in turn have a large economic impact, and those dealing with the ongoing ocean acidification (see above).

# 2. Material and methods

## 2.1. Material

We analysed specimens of the Balanidae *Austromegabalanus psittacus* (subfamily Megabalaninae), sampled alive from Isla Santa María, close to Antofagasta (north Chile) and *Perforatus perforatus* (subfamily Concavinae), collected alive near Almuñécar (southeast Spain). Some specimens of the former species were fixed in 2.5% glutaraldehyde, buffered with cacodylate 0.1 M, pH 7.4, for at least 2 days, and later kept in cacodylate buffer; they were maintained at 4°C during the whole process. For comparison, we also examined specimens of *Notobalanus flosculus* (Archaeobalanidae, Balanoidea), collected dead in Curiñanco (central Chile), and *Austrominius modestus* (Tetraclitoidea, Austrobalanidae), collected alive in Navidad (central Chile).

## 2.2. Optical microscopy

Two specimens of *A. psittacus* were embedded in methacrylate at the Andalusian Center of Nanomedicine and Biotechnology (BIONAND), Málaga, Spain. The samples were dehydrated and embedded in Technovit 7200 VLC in five steps. The first three steps were mixtures of ethanol (ET) and Technovit (T) (30T : 70ET; 50T : 50ET; 70T : 30ET) and the last two steps consisted only of Technovit 7200 VLC. The samples were subsequently polymerized. The embedded tissues were sectioned to a thickness of 50 µm using the cutting band system EXAKT 300 CL. One of the specimens was sectioned perpendicular to the growth axis, while the other was sectioned parallel to this same axis. The sections were ground with a precision micro-grinding system EXAKT 400 C, stained with toluidine blue and photographed with an Olympus VF120 microscope at the University of Málaga, Spain.

## 2.3. Scanning electron microscopy

Ultrasonicated fractures (cleaned with commercial bleach, approx. 5% active chlorine, for 4–5 min), as well as polished sections (previously etched for 2–3 min with 4% EDTA), were prepared for SEM observation. All samples were carbon-coated (Emitech K975X carbon evaporator) and observed in the field emission SEM (FESEM) equipments Zeiss Auriga and FEI QemScan 650 F of the Center for Scientific Instrumentation (CIC) of the University of Granada (UGR), Spain.

## 2.4. Transmission electron microscopy

Pieces of previously fixed specimens of *A. psittacus*, as well as of dry specimens, were completely decalcified by full immersion in 2% EDTA for about two weeks, post-fixed in $OsO_4$ (2%) for 2 h at 4°C and embedded in epoxy resin Epon 812 (Electron Microscopy Science, EMS). Sections were obtained with an ultramicrotome Leica Ultracut R. Semi-thin sections (approx. 0.5 µm) were stained with 1% toluidine blue and observed and photographed with an Olympus BX51 microscope. Ultra-thin sections (50 nm) were stained with uranyl acetate (1%), followed by lead citrate. They were later carbon-coated and observed with TEM (Zeiss Libra 120Plus). All the instrumentation is housed at the CIC (UGR).

## 2.5. Micro-computerized tomography (micro-CT)

Samples were scanned with a ZEISS Xradia 510 VERSA micro-tomograph. A general scan of an individual of *A. psittacus*, around 12 mm in dorsoventral length, was carried out at 80 kV, power value 7.02 W, pixel size 12.2169 µm, optical magnification 0.4× and exposure time 6 s. Another conspecific specimen (31 mm in dorsoventral length) was scanned at 70 kV, power value 6 W, 18.004 pixel size, optical magnification 0.4× and exposure time 3 s. A total of 2034 images were taken in both cases. Image reconstruction was done with Reconstructor Scout and Scan (12.0.8059). Dragonfly Pro (Object Research System) was used for advanced post-processing and quantification of image data for material characterization.

## 2.6. Fourier transformed infrared spectroscopy

The composition of organic membranes was determined by analysing their surfaces using attenuated total reflectance–Fourier transformed infrared spectroscopy (ATR-FTIR). The analyses were performed

with a Jasco 6200 spectrometer, with a total of 32 scans per measure and a resolution of $2\ cm^{-1}$ in the mid-infrared region (4000–400 $cm^{-1}$). The relative percentages of protein, polysaccharides and lipids were estimated from the relative intensities of absorption peak areas associated with the characteristic molecular group of each component (C–H: lipids or fatty acids; amide I and II: proteins; COC: sugars/polysaccharides; see [23] for details).

# 3. Results

## 3.1. Main skeletal elements and growth mode

The shell of the two species of Balanidae studied consists of six wall plates (one rostrum, two rostromarginals, two carinomarginals and one carina) and a basal plate (figure 1a,b; electronic supplementary material, figures S1, S2a–c and video S1). Only the rostrum has two radii, whereas both the rostromarginal and carinomarginal plates have one radius projecting in the direction of the carina (figure 1a–c; electronic supplementary material, figure S2b). The radii and parietes of the rostromarginal plates are somewhat wider than those of the carinomarginal plates (figure 1a,b; electronic supplementary material, figures S1a,b and S2b). At the shell interior, the sheath of the carina is symmetrical and emits two alae towards the carinomarginal plates (figure 1b; electronic supplementary material, figure S2b). The somewhat asymmetrical sheath of each of these plates has in turn one ala projecting towards the rostromarginal plates (figure 1b,d; electronic supplementary material, figure S2b). The ala is accommodated into an articular surface of the neighbouring plate (figure 1d). The same applies to the rostromarginal plates with respect to the rostrum, as the rostral sheath accommodates the alae projecting from each of the two rostromarginal plates (figure 1b; electronic supplementary material, figure S2b). In both *A. psittacus* and *P. perforatus*, the rostral margins of the alae fit into a longitudinal abutment present on the internal surface of the opposing plates [19].

The horizontal distribution of external growth lines demonstrates that the parietes grow at their contact areas with the base plate (figure 1a,c; electronic supplementary material, figures S1 and S2a). These contacts progressively expand laterally with the growth in height of the parietes, which is how they acquire their pseudo-triangular external aspect. The radii are imprinted with very faint dorsal–ventral growth lines, i.e. parallel to their sutural contacts with the adjacent plates (figure 1c), indicating that they grow perpendicular to these contacts. Lateral growth at these margins is inhibited by the settlement of younger individuals, a fact particularly common in *A. psittacus* (electronic supplementary material, figure S1b,c). On the shell interior, the sheath of each plate is characterized by growth lines which are parallel to its dorsal edge (i.e. it grows in the dorsal direction; figure 1d). All alae display growth lines parallel to their dorsoventral growth margins, which imply lateral growth towards the rostrum (figure 1d). In this way, radii growth towards the carina is compensated by alae growth towards the rostrum.

The longitudinal canals of the parietes begin at the base of the parietes as wide spaces between the inner and outer laminae and the primary septa (figure 2a). The growth lines observed in cross sections of the plates provide evidence that, with plate growth towards the dorsum, new material is added to the interior of all of these structures such that the initially wide longitudinal canals begin to narrow and acquire their typical tubular morphology (figure 2b). During this process, more material is added on the outer lamina side of the tubes than on the inner lamina side, in such a way that narrowing of the longitudinal canals is asymmetric (figure 2b). A similar distribution of growth lines around canals was found in the tetraclitid *Tesseropora rossea* [13]. The non-denticulate outer portions of the primary septa coarsen and become the walls (longitudinal septa) between canals (electronic supplementary material, figure S2d,e), while their denticulate inner portions become embedded within the inner lamina as interlaminate figures [16] (figure 2b). Similarly, the secondary septa and the basal teeth extending from the outer lamina become intralaminate figures (figure 2b). A thin layer of translucent material (in thin section; the so-called internal layer [24,25]) begins to be added to the interior of the inner lamina (figure 2b–d), which transforms into the sheath–alae complex in the ventral direction (electronic supplementary material, figure S3). The radii start growing in concomitance with the initiation of the first deposits onto the laminae and septa (figure 2c). All these structures contain internal organic layers (figure 2d), with a conspicuous fibrous nature (figure 2e). These are periodically distributed strictly parallel to the growth lines (figure 2d,f; electronic supplementary material, figure S3), i.e. they mark growth episodes of unknown nature. They are relatively continuous between structures and

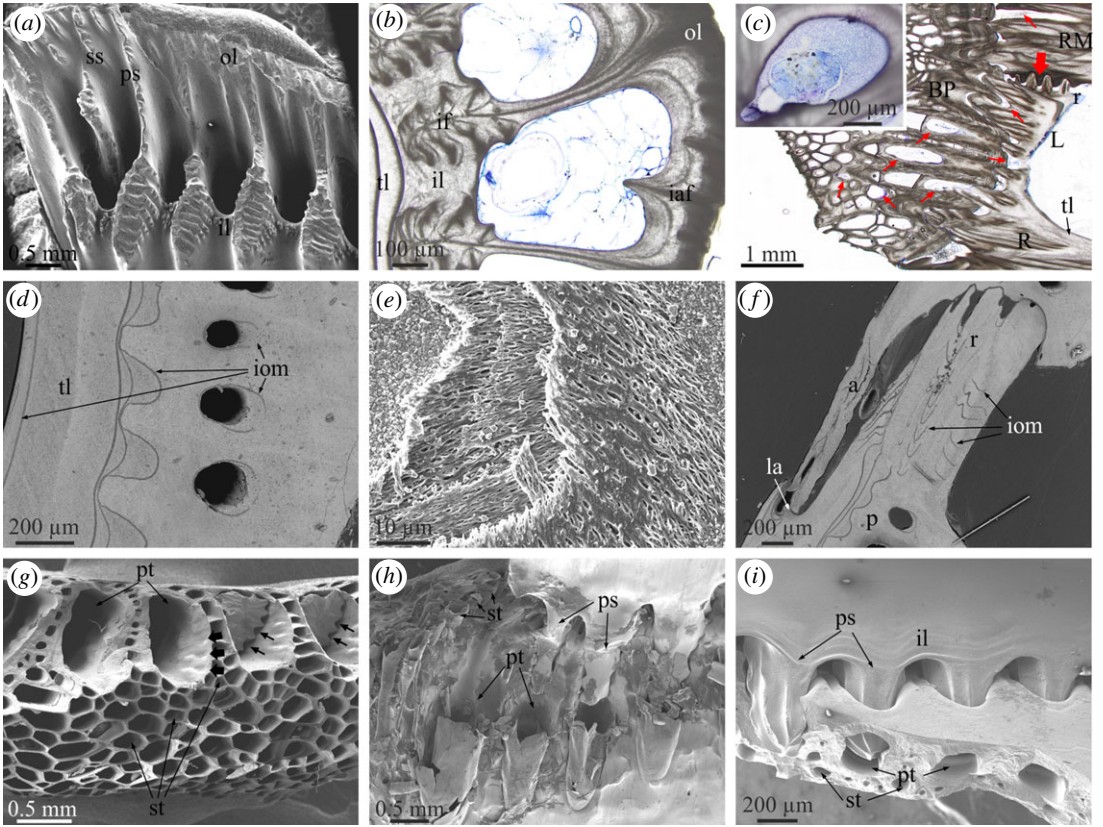

**Figure 2.** Main morphological elements in the Balanidae. (*a*) View of the basal growth margin of a paries of *P. perforatus*. (*b*) Transversal section through a paries of *A. psittacus*. Growth lines indicate growth of the outer lamina towards the interior. The inner lamina coarsens both towards the interior and the exterior and incorporates the denticulate parts of the primary septa (see panel (*a*)) as interlaminate figures. Note cellular material within the canals. (*c*) Longitudinal view of the base and wall plates of *A. psittacus*. The thin red arrows point to sectioned tubes of the base plate and other void structures at the contact of the base plate with the wall plates, which contain remains of cellular tissue. The thick red arrow points to the initiation of a radius. The inset is a transversal section through a longitudinal canal showing the aspect of the infilling tissue. (*d*) View of a horizontally sectioned paries of *P. perforatus*, showing internal organic membranes. (*e*) Fibrous membrane within a lateral plate of *P. perforatus*, partly exhumed by slight decalcification. (*f*) Horizontal section through the contact between plates of *P. perforatus*, showing the distribution of internal membranes; they are particularly patchy in the radius region. (*g*) Fracture of the margin of the base plate of *A. psittacus*, showing the difference between the primary and subsidiary tubes. Note the small grooves accommodating the denticles of the primary septa at the initiation of the primary tubes (thick arrows). Remains of two such septa show the existence of permanent interstices between the denticles and the primary tube wall (thin arrows). (*h*) Partly decalcified specimen of *A. psittacus*, showing the interlocking between the primary septa of the paries base and the ends of the primary tubes of the base plates. Decalcification reveals the organic linings of the primary and subsidiary tubes. (*i*) Interlocking between the denticulate portions of the primary septa and the primary tubes in *A. psittacus*. Note permanent gaps between the inner lamina and the basal plate. (*a*), and (*d*) to (*i*) are SEM views; (*b*) and (*c*) are optical microscope views. a, ala; BP, basal plate; CM, carinomarginal plate; iaf, intralaminate figure; if, interlaminate figure; il, inner lamina; iom, internal organic membrane; la, longitudinal abutment; ol, outer lamina; p, paries; ps, primary septum; pt, primary tube; R, rostrum; r, radius; RM, rostromarginal plate; ss, secondary septum; st, subsidiary tube; tl, translucent layer.

only become patchy within the radii (figure 2*f*). In longitudinal cross sections of the plates, these membranes adapt to the shape of the sheath by curving backwards towards the basal side of the plates (electronic supplementary material, figure S3). Occasionally, some such membranes may be seen around the longitudinal canals (figure 2*d*).

The base plate is characterized by more or less concentric growth lines in dorsal view (figure 1*a*; electronic supplementary material, figure S1*c*), which demonstrate that it grows radially. The plate consists of an internal layer of primary tubes which grow radially from a growth apex towards the periphery (electronic supplementary material, figure S2*c* and video S1). They are sometimes separated by small tubes (figure 2*g*). With growth of the basal plate, the primary tubes seem to grow in diameter. At the same time, the splitting of some tubes introduces new elements (electronic

supplementary material, figure S2c). There is an external layer (close to the substrate) of small subsidiary tubes with polygonal outlines (figure 2c,g–i; electronic supplementary material, figures S2d,e and S3), forming a kind of cellular material. They are sometimes interrupted by thin septa (electronic supplementary material, figure S2d). The extent of the external layer is variable, but it is particularly well developed in A. psittacus (e.g. figure 2g), and reduced or even non-existent in P. perforatus. The peripheral openings of the big tubes are the places where the toothed portions of the primary septa of the parietes fit (figure 2g–i; electronic supplementary material, figure S2f).

The growth apex is not always at the centre of the basal plate. It may be displaced in any direction, even close to the periphery (figure 1a; electronic supplementary material, figures S1c and S2c). This usually happens in crowded conditions, in which the animals tend to grow away from the impinging neighbours (electronic supplementary material, figure S1c). Under impingement with other neighbours, the base plates of the animals extend vertically at the expense of the wall plates (electronic supplementary material, figure S1a,b). In specimens of A. psittacus completely surrounded by other neighbours, the base plate may acquire a conical or tubular shape, sometimes becoming higher than the rest of the shell (electronic supplementary material, figure S1b).

Thin sections of methacrylate-embedded specimens show that not only the longitudinal canals of the wall plates, but also the primary tubes and some subsidiary tubes of the basal plates, as well as the connections between the wall and basal plates, are filled with living tissue (figure 2b,c; electronic supplementary material, figure S3). In some longitudinal canals close to the base, the soft tissue seems to be mantle parenchyma (figure 2c inset).

## 3.2. Lateral growth margins of the plates

### 3.2.1. Radius–paries contacts

In both A. psittacus and P. perforatus, the growth (carinal) margins of the radii are characteristically ornamented with a series of horizontally elongated and evenly spaced crenulae, separated by troughs (figure 3a). The abutting rostral margin of the adjacent plate paries also develops a similar series of crenulae and troughs (figure 3b), which accommodate the complementary elements of the abutting radius margin. The two margins thus constitute an interlocking system. There is nevertheless a fundamental difference between both margins. The crests of the carinal margins of the radii are much more prominent (figure 3c), such that while they are in direct contact with the troughs of the rostral margin of the opposing plate, the crests of the rostral margin of the paries do not meet the troughs of the carinal margin of the radius. Due to this, there are permanent voids between the troughs of the radius margins and the crests of the abutting paries margin (figure 3d). With growth, the subsequently secreted crenulae and troughs of both margins widen (figure 3a,b). In detail, the crenulations of both margins have dendritic shapes, which are particularly conspicuous in the carinal margins of the radii (figure 3a,b). In relatively grown crenulae, it is easy to observe how dendrites become increasingly complicated (i.e. new divisions into new high-order branches appear) towards the interior, i.e. in the direction where the radius thickens (figure 3e,f). Under high magnification, the process of dendrification can be observed beginning within the nascent crenulae, where the smooth crests margins become interrupted by low-relief areas with dendritic outlines (figure 3g,h). The top edges of the crenulations of the radius margins are distinctly smooth (figure 3e inset, g,i) and crystal size is much less (less than 1 μm) than the usual grain size (between 1 and 3 μm; figure 3h). At the opposing rostral margin, not only the crests, but also the troughs and the intervening areas are also smooth, and there are no apparent differences in grain size (figure 3j).

Transversal sections reveal that the crests of the two opposing margins display growth lines parallel to the crest outlines (figures 2f and 3d,k). The growth lines are present throughout the entire radius extension (figures 1b and 3k), while they are restricted to only the teeth interior in the rostral margin of the opposite paries (figure 3d). From an external side view, lateral growth of the crenulations leaves traces on the exterior of the radii, which can easily be confused with horizontal growth lines (figure 1c; electronic supplementary material, figure S2a); however, the actual growth lines are parallel to the vertical radius margin (figure 1c).

The interlocking crest-and-trough system of both margins is not directly in contact. There is always an intermediate membrane (figure 3b inset, c,j–l), which, upon separation of the plates, becomes unavoidably attached to the rostral margin (figure 3b,c,l). It has a smooth aspect under the SEM (figure 3j,l), unlike the fibrous organic membranes which become embedded within the plates (figure 2e). TEM sections of decalcified specimens of A. psittacus show that these intermediate

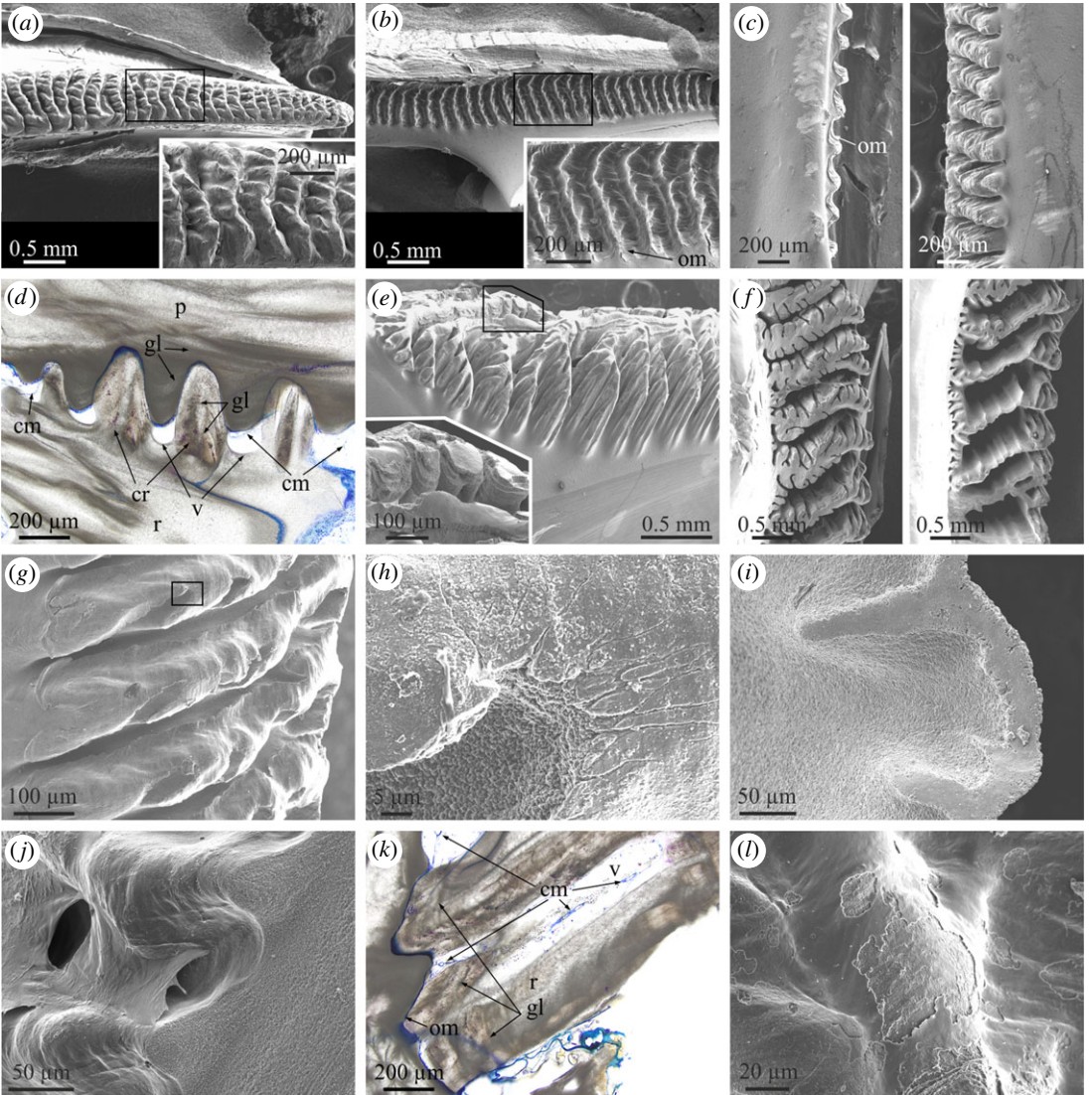

**Figure 3.** Structural elements found at the lateral boundaries between plates. (*a*) View of the carinal margin of the radius of a wall plate of *P. perforatus*. The inset is a detail of the crenulations, showing their dendritic margins. (*b*) Rostral margin of the paries directly opposing the specimen in (*a*). The inset shows the system of crests and troughs covered by an organic membrane. (*c*) Profile views of the rostral margin of the paries (left) and the opposing carinal margin of the radius (right) in *P. perforatus*. Note the membrane covering the paries margin and the much higher elevation of the crenulations of the radius margin. (*d*) Cross-sectional view of the contact between the carinal margin of the radius and the rostral margin of the paries in *A. psittacus*. There are permanent voids between the troughs of the former margin and the crenulations of the latter. Some of them show remains of cellular material. The crenulations of both margins display internal growth lines. (*e*) Aspect of the carinal margin of the radius of a wall plate of *A. psittacus*, close to the plate base. Note the particularly dendritic aspect of the crenulations. The division into lower-order branches takes place towards the plate interior. The inset shows the smooth texture of the crenulation surfaces. (*f*) Different designs of carinal margins of radii of two specimens of *A. psittacus*. The division into new high-order branches takes places towards the shell interior (towards the right of images). (*g*) Incipient crenulations in *A. psittacus*. Dendrites begin to form within each crenulation with the development of gully-like structures. The texture of the top areas of the crenulations is particularly smooth. (*h*) Detail of the area framed in (*g*), showing the difference in texture and grain size between the top area of the crenulation and the interior of one of the gullies. (*i*) Detail of the edge of a crenulation of *A. psittacus*, showing the difference in texture between the flat top and the slope. (*j*) System of crests and troughs of the rostral margin of the paries of *A. psittacus*. Their surfaces are smooth and the intervening intermediate membrane is partly detached from the surface. (*k*) Horizontal section across the contact between the radius and paries in *A. psittacus*. The radius bears internal growth lines. The intermediate organic membrane is stained in blue. Note the existence of cellular tissue filling in the voids left at the troughs of the crenulations of the radius margin. (*l*) View of an area similar to that in (*j*), in *P. perforatus*. The intermediate organic membrane is partly abraded; however, its smooth texture is still evident. (*a*) to (*c*), (*e*) to (*j*) and (*l*) are SEM views; (*d*) and (*k*) are optical microscope views. cm, cellular material; cr, crenulation; gl, growth line; om, organic membrane; p, paries; r, radius; v, void.

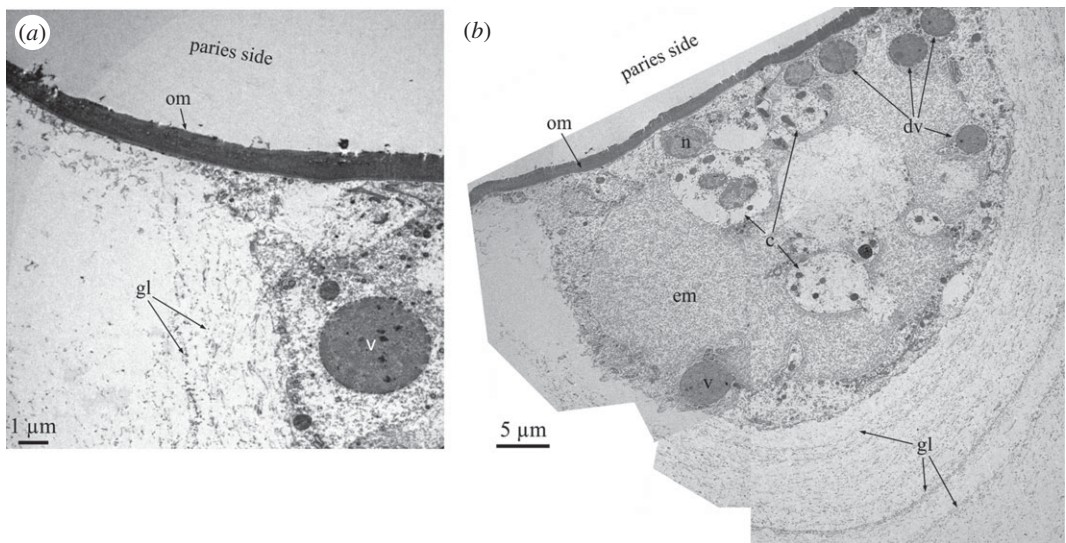

**Figure 4.** TEM analysis of the membranes observed at the radius–paries boundaries and associated cellular material. (*a*) View of the organic membrane between the carinal margin of the radius and the rostral margin of the paries of *A. psittacus*. (*b*) Composite micrograph of the cellular material found at the trough between crenulations of the carinal margin of the radius of the wall plate of *A. psittacus* (cf. figure 3*d*). Upon decalcification, the growth lines are revealed by aligned fibrils. c, cell; dv, v, dense vesicles; em, extracellular matrix; gl, growth lines; n, cell nucleus; om, organic membrane.

membranes are electron-dense and weakly laminated (figure 4*a*; electronic supplementary material, figure S4*c*). Measured thicknesses range between 1.5 and 5 µm. TEM analysis also reveals that the voids between the crests of both margins are infilled with cellular material (figure 4*b*); this is also visible in methacrylate-embedded and sectioned specimens (figure 3*d*,*k*). It is also to note the low number of cells present in these organic pockets (figure 4*b*). These cells secrete large dense vesicles to the extensive fibrous extracellular matrix (figure 4*a*,*b*). Under TEM, the decalcified growth lines appear as tiny and diffuse laminae constituted by fibrils with nanometric thicknesses (figure 4*b*).

### 3.2.2. Ala–sheath contacts

The growth edges of the alae fit snuggly in a longitudinal ridge (longitudinal abutment) present at the margin of the adjacent plate's sheath (figure 5*a*, *a* inset, *b*). In a way similar to the radius–paries contacts, the edges of the alae are also crenulated and display some degree of dendrification. However, due to their reduced width, dendrification is much less than in the case of the radii edges (figure 5*a*,*c*,*d*). Similarly, branching occurs in the thickening direction, i.e. towards the shell interior (figure 5*c*,*d*). The opposed longitudinal abutment is parallel to the edge of the sheath (figure 5*e*,*f*). The crenulations of the alar surface adapt to much less prominent features along the base of the longitudinal abutment (figure 5*f* inset).

Between the ala and the longitudinal abutment, there is an organic membrane (figure 5*b*), which is particularly thick (approx. 2–5 µm in *P. perforatus* and approx. 10 µm in *A. psittacus*; figure 5*g*). In a specimen of *P. perforatus*, the surface of the membrane looking towards the longitudinal abutment is imprinted with growth lines (figure 5*g* inset). The adhesion of this membrane to the smooth longitudinal abutment surface does not seem to be as permanent as in the case of the rostral margin of the paries and frequently the dry membrane detaches from the longitudinal abutment and folds over the alar margin (figure 5*c*,*g*). The top surfaces of the crenulated margin of the alae, i.e. those in contact with the membrane, have a much smoother surface texture than the depressed zones (figure 5*c*,*d*,*g*) and are particularly fine-grained (figure 5*d*,*g*). We have not been able to observe cellular material at these margins. At the point where the margin of the ala ceases to be in contact with the longitudinal abutment in the ventral direction, the crenulations disappear and the margin becomes smooth, or, at most, slightly undulated in coincidence with the growth lines (figure 5*h*). The organic membrane is also persistent in this contact-free zone.

A very wide scar extends from the abutment over the radius' internal surface (figure 5*e*,*f*). It can be identified as the articular surface for the adjacent ala [21]. Its dorsal margin is more or less at the level of the sheath and, towards the carina, its margin changes and is both parallel and close to the radius margin (figure 5*f*). This scar has a much smoother surface texture than the adjacent internal surface of the radius

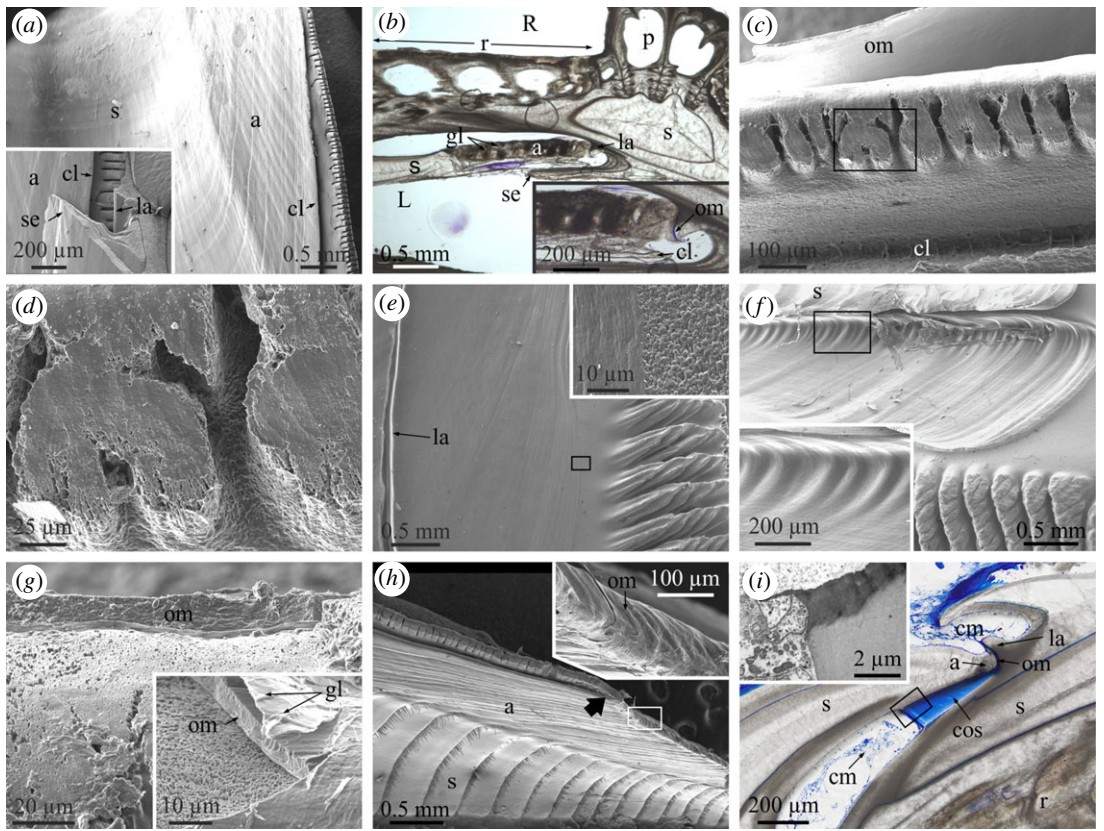

**Figure 5.** Structural elements found at the contacts between the alae and sheaths. (*a*) General view of the sheath–ala complex of *A. psittacus*. The crenulated margin of the ala is separated by a cleft. The inset is a fracture through the ala and the sheath of the opposite plate, showing the mode of insertion of the ala into the lateral abutment. (*b*) Horizontal cross section through the junction between the ala of the wall plate and the sheath of the rostrum in *A. psittacus*. The ala and sheath of the wall plate can easily be differentiated by the presence of growth lines in the ala. The inset is a detail of the junction, in which the intermediate organic membrane (stained in blue) can be discerned. (*c*) Crenulated margin of the ala in *A. psittacus*. Note the incipiently dendritic aspect of the crenulations. The intermediate membrane can be seen at the top. The cleft separating the crenulated edge from the rest of the ala is at the bottom. (*d*) Detail of the area framed in (*c*). The top areas of the dendrites have a smooth surface texture. (*e*) View of the articular surface for the adjacent ala and the radius edge of a wall plate of *A. psittacus*. The inset is a detail of the framed area and shows the difference in surface texture between the articular surface and the rest of the internal surface of the plate. (*f*) Similar view in *P. perforatus*. The inset is a detail of the framed area and shows the set of smooth crests and troughs of the base of the longitudinal abutment, in which the edge of the ala fits. (*g*) Aspect of the organic membrane covering the ala edge in *A. psittacus*. The inset is a detail of the same type of membrane in *P. perforatus*. Its surface is imprinted with growth lines. Note the fine-grained texture of the crenulation surface. (*h*) Aspect of the sheath–ala complex in *P. perforatus*. The arrow marks the boundary between the denticulate and non-denticulate sections of the edge of the ala. The inset is a view of the non-denticulate part of the edge of the ala (framed area). (*i*) Horizontal section through the contact of ala and sheath in *A. psittacus*. The intermediate membrane between the edge of the ala and the longitudinal abutment appears stained in blue. This membrane continues into a thick and dense laminated connective organic structure, which extends in the direction towards the radius edge. The cellular material responsible for the secretion of this thick membrane is indicated. The inset is a TEM view of the contact between the connective organic structure and the secreting cells, taken from an area similar to that framed in the optical micrograph. (*a*) and (*c*) to (*h*) are SEM views; (*b*) and (*i*) are optical microscope views. a, ala; cl, cleft of the alar margin; cm, cellular material; cos, connective organic structure; gl, growth line; la, longitudinal abutment; om, organic membrane; p, paries; R, rostrum; r, radius; RM, rostromarginal plate; s, sheath; se, sheath extension.

(figure 5*e* inset), and is imprinted with growth lines (figure 5*e*,*f*), which implies marginal growth. In horizontally sectioned methacrylate-embedded *A. psittacus*, a laminated dense organic structure extends from the base of the longitudinal abutment, as a continuation of the intermediate membrane, onto the articular surface for the adjacent ala (figure 5*i*). This structure connects the ala and the radius of opposite plates. Accordingly, we call this the connective organic structure. Its laminae are parallel to the growth surface, which is found in the direction of the carina and directly in contact with the cellular material responsible for its secretion (figure 5*i*, *i* inset).

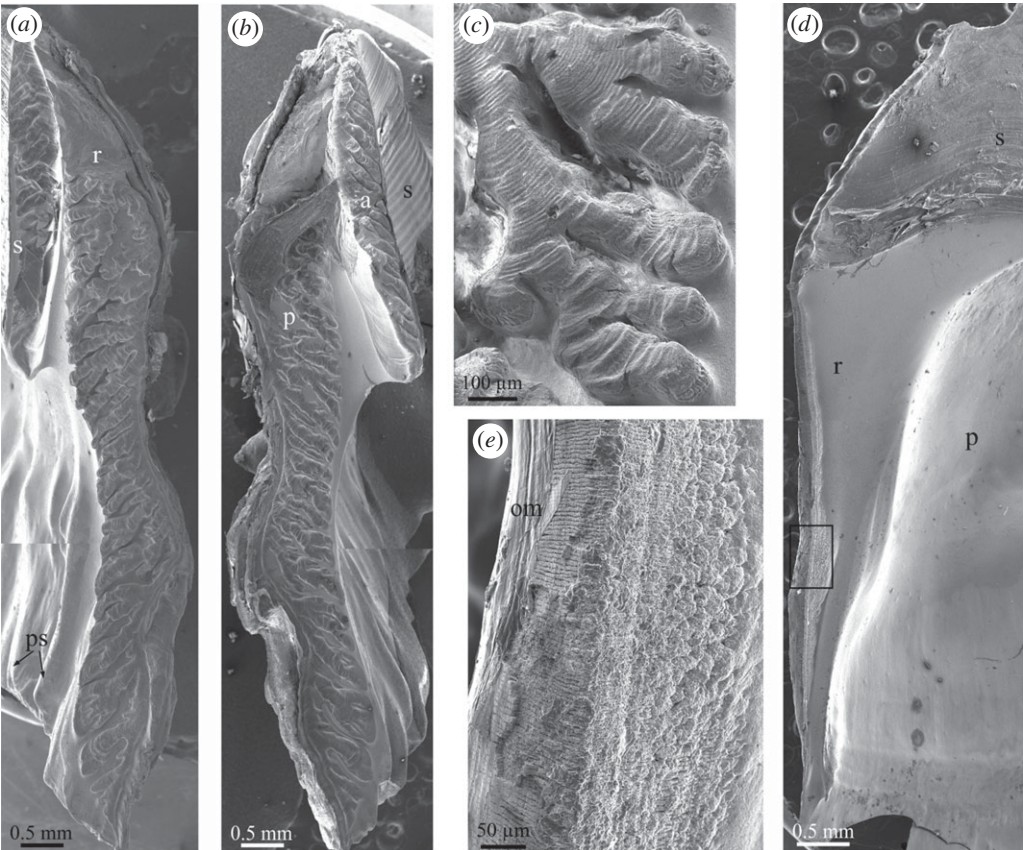

**Figure 6.** Wall plate boundaries of the studied non-balanid species. (*a*) General view of the carinal edge of a wall plate of *N. flosculus*. (*b*) Margin of the carinomarginal plate abutting the carinal edge seen in (*a*). (*c*) Detail of the margin of a radius similar to that in (*a*). Growth lines are clearly imprinted on the high-relief dendritic features. (*d*) Carinal edge of a wall plate of *A. modestus*, showing the absence of dendritic features. (*e*) Detail of the area framed in (*d*). Note microscopic ridges at the abutting margin, as well as the intermediate organic membrane. All SEM views. a, ala; p, paries; om, organic membrane; ps, primary septum; r, radius; s, sheath.

### 3.2.3. Non-balanid species

In the small-sized archaeobalanid *Notobalanus flosculus*, there are extensive contact areas between the radii and the parietes. The alae directly abut the carinal margins of the sheaths, also at extensive contact areas. All these surfaces display very characteristic dendritic features, which, contrary to the balanids examined, are not organized into crests. The dendritic patterns found at both types of growth margins are strikingly similar. At the carinal margins of the radii and at the rostral margins of the alae the dendrites are in high relief. The contrary is true for the abutting paries and sheath margins (figure 6*a*,*b*). Despite the material being collected dead, the intermediate membranes were still present and preferentially adhered to the paries and sheath margins. The conspicuous growth lines associated with the dendrites indicate preferential growth towards both the animal's interior and dorsum (figure 6*c*). The dendrites clearly branch in the growth direction.

No similar dendritic structures have been observed in the tetraclitoidean *A. modestus*. The radii and parietes, and the alae and sheaths mostly overlap laterally and only abut with each other along very reduced marginal areas (100–200 µm thick; figure 6*d*). No apparent crenulations or dendrites are present. The calcite grains forming the shell are arranged in tiny ridges (4–5 µm wide), which may bifurcate interiorly (figure 6*e*). Given their size and arrangement, they may be compared to growth lines. Very thin intermediate organic membranes have been observed (figure 6*e*).

## 3.3. Basal growth margins

In the two balanid species studied, the basal margins of the plates also interlock with the marginal area of the base plate mainly by means of radially arranged primary septa (figures 2*a* and 7*a*,*b*). There are short secondary septa which extend from the outer lamina (figures 2*a*,*b* and 7*a*,*b*). The primary septa become

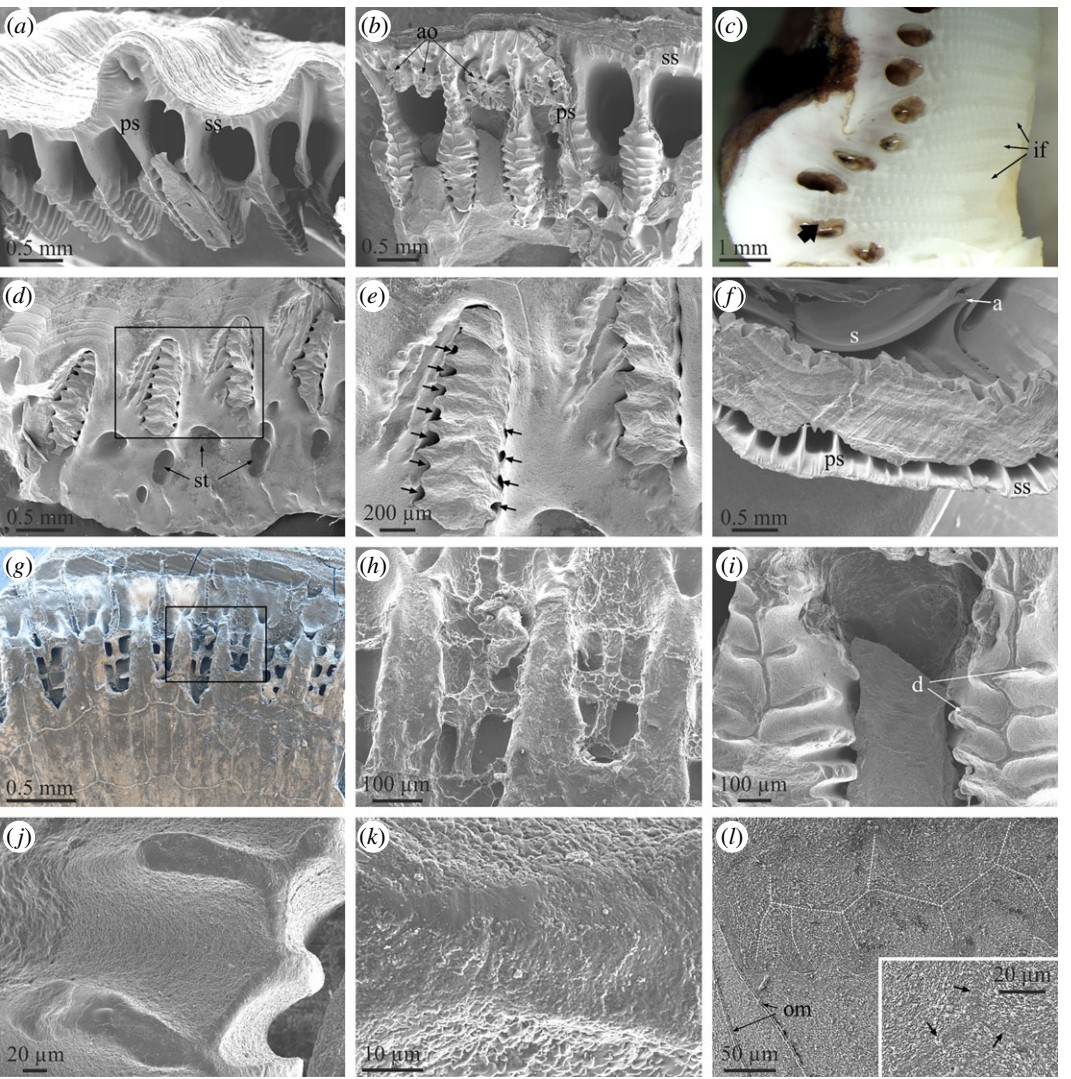

**Figure 7.** Structural elements observed at the contacts between the wall plates and the basal plate. (*a*) View of the base of a rostral plate of *A. psittacus*, showing the system of primary septa and canals. (*b*) Similar view of a wall plate of *P. perforatus*. Note the presence of abnormal outgrowths with dendritic outlines. (*c*) Polished section of the contact between the rostromarginal and carinomarginal plates of *A. psittacus*. The interlaminate figures are particularly extensive and there is one (arrow) which is denticulate all along its extension. (*d*) Peripheral area of the base plate of *P. perforatus*. Remains of the denticulate parts of the primary septa, broken off upon detachment, remained in their original position at the ends of the primary tubes of the base plate. (*e*) Detail of the area framed in (*d*). The arrows point to permanent interstices between the denticles of the primary septa and the walls of the primary tubes. (*f*) View of the contact between the basal plate and carinomarginal plate in *A. psittacus*. (*g*) Peripheral area of the basal plate of a juvenile *A. psittacus*. Upon drying and detachment, the organic membranes at the ends of the primary tubes of the basal plate replicated the outlines of the primary septa, including the denticles, which were formerly in contact with them. The back scatter mode allows us to discern the areas which contain calcite (in white) from those purely organic (in dark hue). (*h*) Detail of the area framed in (*g*) showing the imprints of the denticles of the primary septa. (*i*) Aspect of the denticulate parts of the primary septa in *P. perforatus*. The edges of the septa and denticles have a different texture from the rest of the septa. (*j*) Detail of the edges of denticles of primary septa. Here, their smooth texture can be appreciated. (*k*) Close-up view of one such edge. Apart from the smoother surface texture, the grain size is much smaller than that of the intermediate areas. (*l*) Polished and etched section through an interlaminate figure in *P. perforatus*. The branches have been outlined with white broken lines. The inset shows that the branches are made of particularly small grains; arrows point to branches. All SEM views, except for (*c*) (optical micrograph, reflected light). a, ala; ao, abnormal outgrowth; d, denticles; if, interlaminate figure; ps, primary septum; s, sheath; ss, secondary septum; st, subsidiary tube.

branched interiorly through the development of a series of transversal teeth or denticles. In *P. perforatus*, the outer, non-denticulate portion of the primary septa are sinuous and, sometimes, incipiently branched (figure 7*b*), while in *A. psittacus* they are straight (figure 7*a*). In *A. psittacus*, some interlaminate figures

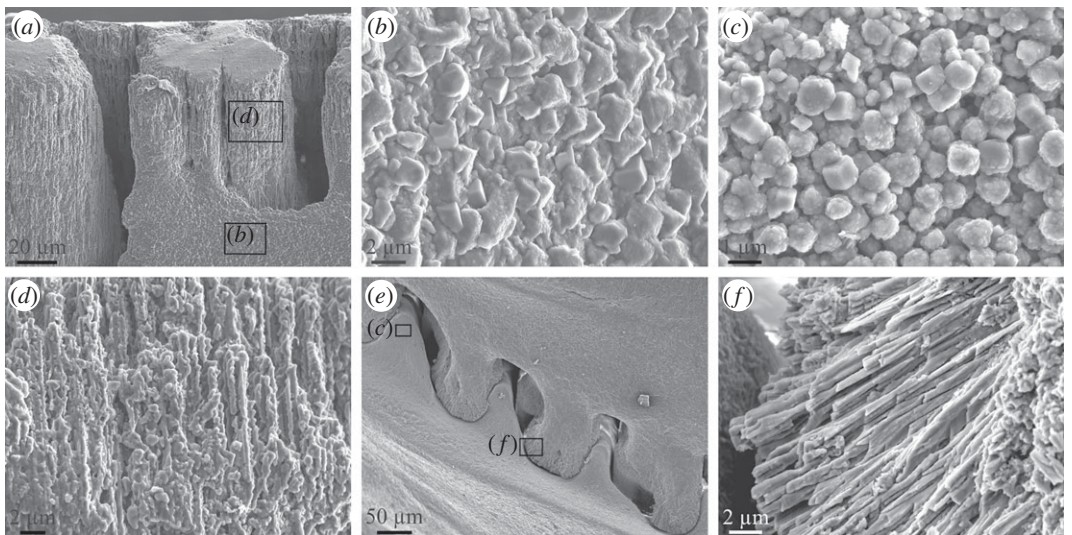

**Figure 8.** Microstructures observed in *A. psittacus*. (*a*) View of the alar margin of a rostromarginal plate. Whereas the crenulations are made of calcite fibres, the shell interior has a granular rhombohedral microstructure. (*b*) Calcite granules observed in the wall of the ala shown in (*a*) (framed area). (*c*) Calcite granules (mostly rhombohedra) observed in the internal surface of a crenulation of the paries shown in (*e*) (framed area). (*d*) Close-up view of the fibrous microstructure of the alar crenulation shown in (*a*) (framed area). (*e*) View of the interlocking system of the crenulations of a radius (top of the image) and the opposing paries (bottom of image). Note the fibrous nature of the radius crenulations (*f*) and the granular nature of the paries crenulations (*c*). (*f*) Detail of the fibres observed within the radius crenulation shown in (*e*) (framed area). All SEM views.

almost reach the outer lamina, which implies that the primary septa were entirely denticulate (figure 7*c*). Some abnormal outgrowths within the primary canals of *P. perforatus* are also clearly dendritic in plain view (figure 7*b*). The toothed portion of each septum interlocks with one of the elongated openings of the big radial (primary) tubes of the basal plate (figures 2*g–i* and 7*d*). Accordingly, there is a one-to-one correspondence between the primary tubes of the basal plates and the primary septa of the wall plates (electronic supplementary material, figure S2*c–e*). The denticles of the septa adapt tightly to the periphery of the tube openings. The insertions of denticles are sometimes marked in the interior of the primary tubes as shallow short grooves (figure 2*g*). Nevertheless, there are permanent gaps between the highly prominent denticles and the tube periphery (figures 2*g* and 7*d,e*; electronic supplementary material, figure S2*d–f* and video S1). More exteriorly, the non-denticulate portions of the primary septa, the secondary septa and the primary portion of the longitudinal canals of the wall plates overlap the marginal part of the basal plate, which includes the outer layer of small subsidiary tubes (figure 7*f,g*; electronic supplementary material, video S1).

All canals of the basal plate are internally lined by an organic membrane (figure 2*h*). In specimens in which the wall plates had been abruptly removed, thus exposing the periphery of the basal plate, some dried organic membranes emanating from the primary tube openings showed crests extending from the primary tube interior to the outer lamina. These replicate the edges of the primary septa and their denticles (figure 7*g,h*). Secondary electron images also revealed remains of calcium carbonate belonging to the detached septa (figure 7*g*).

Similar to the sutural edges of radii, the surfaces of the most elevated parts of the septa and denticles are particularly smooth and composed of minute grains, which are markedly smaller than those found in between the elevations (figure 7*i–k*). As the animal grows, the denticulate septal portions become overgrown and are incorporated into the inner lamina as interlaminate figures. In thin section, the branches of the interlaminate figures appear particularly dark (e.g. figure 2*b*). Under the SEM, the same areas are composed of very small grains and are rich in organic matter (figure 7*l*).

The non-balanid species observed (*N. flosculus* and *A. modestus*) have purely organic basal adhesive layers. While ill-defined septa appear in *N. flosculus* (figure 6*a*), the bases of the wall plates of *A. modestus* are devoid of any ornamentation (figure 6*d*).

## 3.4. Microstructures

In both species, the shell is constructed by calcite grains between 0.5 µm and approximately 4 µm, the most frequent sizes being around 1 µm (figure 8*a–c*). Grains may vary in shape from rounded to

euhedral, with rhombohedral morphologies being frequent (figure 8b,c). They have relatively coarse nanogranular textures typical of biominerals. The only markedly different microstructure has been observed very locally at the crenulations of both the radius and ala margins. These consist of fine (0.5–1 µm thick) fibrous elements perpendicular to the growth surfaces, which have serrated edges in the alar margin (figure 8a,d) and straight edges in the radius margin (figure 8e,f). At the growth surfaces, these fibres appear as particularly small grains (described in §3.2.1 and 3.2.2). The low crenulations of the parietal margin opposing the radius are made of rhombohedral grains (figure 8c,e). Clearly, the fibrous microstructure develops only when the growth surfaces are directly in contact with a non-cellular laminated membrane. No similar observations could be made on the basal margins of the primary septa.

## 3.5. Isolated organic membranes

Under the SEM, only the internal membranes appear particularly fibrous (figure 2e), whereas the membranes found at the margins between plates are smooth (figures 2h, 3j,l, 5g and 7h). In TEM section, all membranes are laminated, with laminae spaced, though not evenly, at approximately 100–300 nm (electronic supplementary material, figure S4). Only the membranes observed at the paries–radius margins are more loosely laminated (electronic supplementary material, figure S4c).

The chemical composition of membranes found at the contacts between plates, and within and on top of the plates of the species A. psittacus have been examined by means of ATR-FTIR (figure 9). Membranes were obtained by decalcification of the rostral margin of the parietes (abbreviated PRM; figure 9), the rostral alar margin (RAM), the outer cuticle (Cu), the fibrous membranes internal to the plates (Int), the interior of the primary tubes of the basal plate (PT) and the membranes covering the peripheral portion of the base plate (BPP). Spectra of the different membranes are shown in figure 9a. They all display conspicuous peaks of amide I, II and III groups, from proteins, at 1630, 1520 and 1230 cm$^{-1}$, respectively. Polysaccharides contribute with bands at 1020 cm$^{-1}$, 1060 cm$^{-1}$ and 1160 cm$^{-1}$, from C–O–C, and bands at 2855 and 2925 cm$^{-1}$, from C–H groups. Lipids also contribute to the C–H bands. The disposition and relative intensity of polysaccharide peaks match those of chitin [26,27], thus suggesting that this, together with proteins, is a major component of the membranes.

On the other hand, infrared spectra show considerable differences in the composition of membranes, revealing variations of the relative quantities of chitin and proteins. The normalized intensities of the major chemical components (figure 9b) indicate that the RAM membranes are significantly richer in proteins that the rest, followed by Cu, PRM and Int membranes, with the lowest values corresponding to PT and BPP membranes (ANOVA, $p < 0.05$). The intensities of chitin/polysaccharide peaks were always much smaller than those of proteins, and follow an almost opposite trend to proteins: the BPP membranes are the richest, followed in descending order by Int, PT, PRM, Cu and RAM membranes. The intensities of bands of C-H groups (lipids) are very small; and decreases following the sequence BPP, RAM, PT, Int, PRM, Cu (figure 9b).

# 4. Discussion

Balanid cirripeds are characterized by diametric growth. Growth in height is achieved by growth of the plate parietes at their dorsal (basal) margin, while growth in width of the aperture takes place both at the carinal growth surfaces of the radii and at the rostral growth margins of the alae (figure 1). All radii grow towards the carina, whereas alae grow towards the rostrum. Growth of both kinds of structures compensate for each other (figure 1b).

## 4.1. Non-cellular membranes at the growth margins and the relationship with the calcified structures

Despite extensive studies of the group, very few studies have dealt with the structures existing at the lateral and basal boundaries between plates. The sets of crests and troughs and the organic membranes at the radius–paries and ala–sheath contacts of a few Balanidae and Tetraclitidae were described only in [5,13,22]. Other than this, the nano- and microstructure, extension and many other details were lacking.

All these growth boundaries share the presence of an intermediate membrane between the two plates in contact. In all cases, our FTIR data indicate that the membranes internal to plates of A. psittacus are

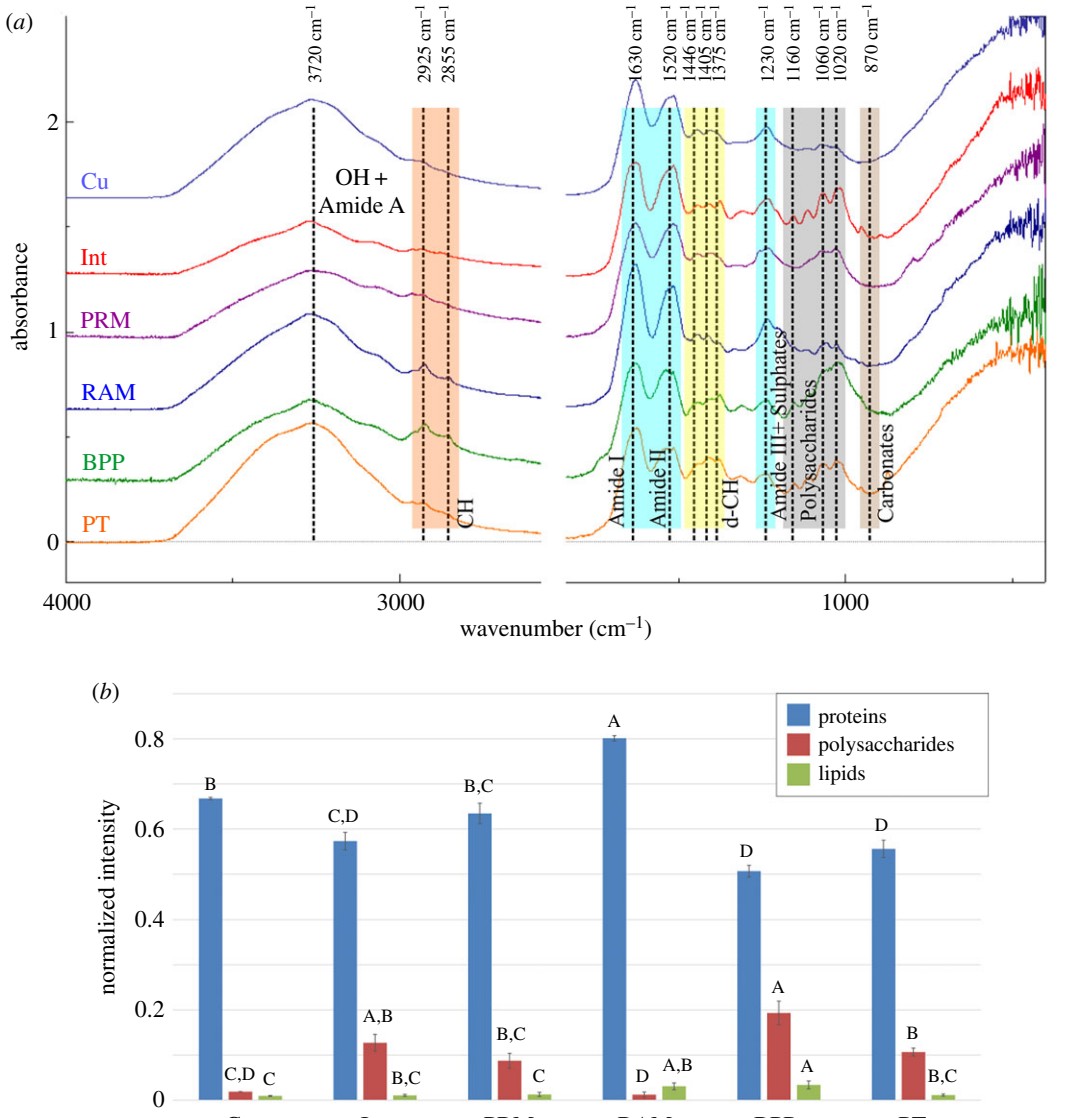

**Figure 9.** (*a*) ATR-FTIR spectra of the different membranes obtained from the shells of *A. psittacus* by decalcification. (*b*) Relative intensities associated with the main organic components: proteins (amide I + amide II), polysaccharides and lipids (CH) of the different membranes; A–D are the groupings obtained with Tukey's method for a 95% confidence for the comparison of proteins, polysaccharides and lipids among the membranes analysed. BPP, peripheral membranes of the base plate; Cu, cuticle; Int, membranes of the lateral plate interiors; PRM, membranes covering the rostral margin of the paries; PT, membranes lining the primary tubes of the plate base; RAM, membranes covering the rostral alar margin.

composed of protein and, to a lesser extent, chitin (figure 9), in accordance with previous results [27,28]. They invariably appear laminated under the TEM (electronic supplementary material, figure S4). This kind of lamination was found in the cuticle of several balanomorphs (including Balanidae) [29,30], and in the membranes internal to the wall plates in *A. psittacus* [28]. The membranes at the contacts of radii and parietes attach to the rostral side of the parietes (figure 3*b,j,l*). The dendritic edges of the crenulations of the carinal margin of the radii are much smoother and consist of much more minute grains than the rest of the crenulation surfaces (figure 3*h,i*). This smooth surface texture implies that they are the only parts of the crenulations in permanent contact with the organic membranes, which is supported by the cross-sectional views in methacrylate-embedded specimens (figure 3*d,k*). This permanent contact may explain the differential smooth surface texture, if the membrane limits the grains' ability to grow and protrude from the growth surface. The contact of each ala with its corresponding longitudinal abutment is directly comparable to the above in that there is an intermediate membrane. The alar margin is made of dendritic crenulations with particularly smooth top surfaces (figure 5*a,c,d,g*). The dendritic aspect changes to smooth right when the alar margin

ceases to be in contact with the longitudinal abutment, although the organic membrane remains continuous (figure 5h). At the third type of contact, that between the wall and basal plates, there is also a membrane which extends from the interior of the primary tubes over the marginal area of the plate base. In some samples, it can be appreciated how the outlines of the primary septa (including denticles), of the bases of the paries are imprinted onto this membrane (figure 7g,h). In a way similar to the other types of plate contacts, the edges of the primary septa are the only structures of the wall plate margins in contact with the membrane, which fits in with their particularly smooth surface texture (figure 7j,k). In addition, these edges consist of particularly small grains and are particularly rich in organic matter. This is shown by the high contrast of the interlaminate figures and the rest of the structures related to the septa in thin sections (figure 2b). All this is also consistent with these structures growing in contact with an organic-rich environment from which they absorb organic matter. In summary, in the three types of growth boundaries, there is permanent contact between the dendritic edges of the parietes, radii and alae, and the intermediate organic membranes, which, except in the latter case, remain attached to the opposing edges.

## 4.2. Growth of plate margins in the presence/absence of living tissue

The existence of mantle cells within the longitudinal canals of the wall plates in *A. psittacus* was previously mentioned [31]. The distribution of soft tissue in our methacrylate-embedded sectioned specimens and the examination of the above contacts reveals that at the wall and basal plates contacts, the animal's soft tissue continues into the canals of both the wall and basal plates (figure 2c; electronic supplementary material, figure S3) across the gaps left by the inner lamina, in between the primary septa (figure 2i). From here, the tissue diverts upwards towards the longitudinal canals of the parietes, and downwards towards (i) the primary canals of the base plate, through the interstices left between the terminal openings of these canals and the denticles of the interlocking inner ends of the primary septa (figure 7d,e), and (ii) the subsidiary tubes of the outer layer of the base plate, through the spaces between the outer lamina and the longitudinal septa (figure 7f). These connections are necessary to enable that all of these structures grow continuously. The tissue observed within the longitudinal canals (figure 2c inset) resembles the mantle parenchyma found in *Amphibalanus amphitrite* [32]. These authors observed that part of the female gonad penetrated the longitudinal canals. At the radius–paries suture, the soft tissue occupies the small gaps left between the troughs inter-crenulations of the radius' carinal surface and the elevations of the paries' rostral surface (figures 3d and 4). Although we do not have direct evidence, a similar situation must take place at the ala-longitudinal abutment contact, where the soft tissue enters the narrow troughs between the elevations (crenulations) of the rostral growth margin of the ala (figure 5c,d). Accordingly, the soft tissue of the animal cannot be in direct contact with all those structures permanently lined by organic membranes: (i) the very edges of the primary septa of the parietes and their denticles, (ii) the ends of the primary tubes and their extensions towards the outer lamina, (iii) the dendritic edges of the crenulations of the carinal surfaces of the radii, (iv) the abutting crenulations and troughs of the rostral margins of the parietes, (v) the dendritic edges of the crenulations of the rostral surfaces of the alae, and (vi) the carinal slopes of the longitudinal abutments. Despite this, the surfaces of these structures are able to actively grow. The only ones that do not seem to grow in thickness are the troughs of the parietes' rostral margins (iv, above) (figure 3d) and the articular surface for the adjacent ala (vi) (figure 5e,f). This implies that the membranes have to allow for the diffusion of the construction materials (whether $Ca^{2+}$ and $CO_3^{2-}$ ions, or calcium carbonate nanoaggregates).

The differences in the microstructure observed between the growth surfaces of the plates directly in contact with the animal epithelium (granular rhombohedral microstructure; figure 8a–c) and those in contact with non-cellular laminated membranes (fibrous microstructure; figure 8d–f) are without a doubt related to the different modes of crystal growth, although it is not presently possible to be more precise in this respect.

What is even more intriguing is how the organic membranes can grow in those areas where the calcified elements abut each other directly, i.e. they are deprived of direct contact with any cellular material. The laminated outer cuticle grows in direct contact with underlying epithelial cells [29,30], but it is unclear how the laminae can be added to the growth surfaces of the membranes which are out of contact with cellular secretory material. The possibility that the cellular material can spread to these areas by periodic separation of the plates, or that organic components (e.g. extracellular secretions; figure 4b) are somehow injected to these areas, seems unlikely. Alternatively, the membranes observed at the radius–paries and ala–sheath contacts might be fully formed very early at

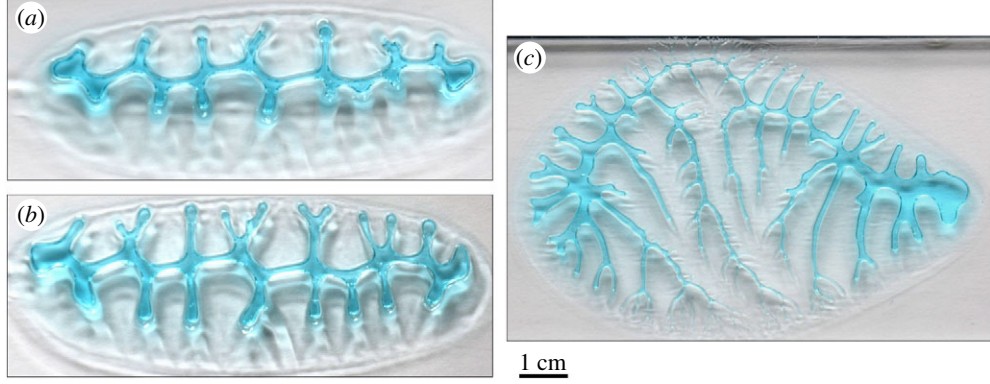

**Figure 10.** Viscous fingering patterns with different complexities formed between a gel (blue) and air. The complexity of the pattern increases from (*a*) to (*c*), by increasing the speed at which air is forced into the gel. The pattern in (*a*) contains only first-order branches, the pattern in (*b*), some second-order branches, while the pattern in (*c*) develops up to fourth-order branches.

the bases of both the wall plates and the ala–sheath contact zone, respectively, and then be transported dorsally with plate growth. Further research is needed to elucidate the mode of growth of these intermediate membranes.

## 4.3. A biophysical model for the formation of dendrites at growth margins

Something which is common to the lateral and basal boundaries between plates is the dendritic aspect of the features developed at these contacts, previously noted only for the sutural edges of the radius of a tetraclitid [13], although these dendritic crenulations were considered as teeth and denticles, following the usual terminology [21]. When looking at all these dendritic features, it becomes clear that the term denticle does not seem adequate, since we are not dealing with discrete features.

These dendritic traits could be interpreted as formed through the action of groups of cells, which shape them following a complex genetically determined behaviour. However, although this explanation is possible, it is unlikely, given the intricate dendritic morphologies and the high diversity of patterns observed (figure 3*a,e,f*). Another possibility that we strongly support here is that they arise through simple physical processes, in particular, through the formation of the so-called viscous fingering [33]. This arises at the contact between two immiscible fluids of different viscosities with the simple condition that the more viscous fluid is pressed by the less viscous fluid. Then, the interface becomes unstable (Saffman–Taylor instability) and forms viscous fingers.

It must be noted that all three kinds of dendritic edges grow attached to the corresponding organic membranes. Accordingly, at the contacts between plates, there are two fluids (not to be confused with liquids) in contact: the highly viscous mucous membranes which are intermediate between plates, and the less viscous cellular and extracellular tissue (e.g. figure 4*b*). It is to stress that Saffman–Taylor instabilities also develop in viscoelastic materials [34], as are most soft biological materials. It has been hypothesized [27] that calcite crystals of the plates of *A. amphitrite* grew within a hydrogel, which could contribute to the low-viscosity fluid in a Saffman–Taylor instability formation scenario. For all biophysical conditions for the formation of viscous fingers to be fulfilled, the pressure at the membranes should be less than at the surrounding cellular tissue. During balanid body growth and in order for a new growth increment to be created, the plates should separate minimally at their edges. This might be achieved by the interplay of increased hydrostatic pressure and muscular tension of the soft body, both mechanisms related to support and feeding behaviour in barnacles (e.g. [35,36]). Since the edges of the dendrites at the different boundaries are attached to the organic membranes, plate separation would involve a certain local pulling of these membranes, which would cause a drop in the internal pressure of the organic viscous phase. Accordingly, this phase would be slightly penetrated by the cellular organic phase. In these conditions, the interface between both organic (cellular + extracellular and non-cellular) materials will develop viscous fingers. Similar elongated dendritic patterns can be produced experimentally, with the lifting version [37]. This is found when viscous fingering occurs between two separating plates (figure 10; electronic supplementary material, figure S5), which is exactly the situation we encounter at the balanid plate boundaries. An additional argument in favour of the viscous-fingering origin of the dendritic features of the growth boundaries

in balanids is the presence of abnormal outgrowths initiating at the outer lamina in *P. perforatus* (figure 7*b*). In our hypothesis, their cross-sectional outlines are also dendritic simply because they were also in contact with the membrane covering the edge of the base plate. The same applies to the non-denticulate portions of the primary septa in the same species (figure 7*b*), although not in *A. psittacus* (figure 7*a*), along the edges of which the contact seemed to be minimal. That the alar margin changes from dendritic to smooth when the contact with the longitudinal abutment is lost (figure 5*h*) is also appealing, since adhesion of the organic membrane to the opposing edge is a necessary condition for the formation of viscous fingers.

As commented on above, the formation of dendritic features reaches its maximum in the radial and alar margins of *N. flosculus* (figure 6*a*,*b*), where no crenulations appear and the dendrites are connected all along the margin. The distribution of growth lines also shows how dendrification progresses with growth of these margins (figure 6*c*).

There are, nevertheless, differences between the margins of the denticulate primary septa and the purely dendritic margins of the crenulations of both the carinal margins of the radii and the rostral margins of the alae. In the first case, new second-order branches (denticles) may be incorporated with time towards the shell interior (compare different growth stages in figures 2*a*,*b* and 7*a*,*b*,*c*,*g*), but these are never observed to subdivide into minor-order branches. On the contrary, the crenulations of the carinal margins of the radii split into third- and even fourth-order branches (figures 3*e*,*f* and 6*a*,*b*) towards the internal shell surface, which provides them their typical dendritic morphologies. Owing to their limited width, the rostral margins of the alae only develop second-order branches towards the shell interior (figure 5*c*,*d*). Although both morphologies (with and without further branching) can be replicated experimentally (by, e.g. changing the pull-out velocity of the plates; compare figure 10*a*,*b* with figure 10*c*), here we cannot provide a straightforward explanation for the differences observed between septa and crenulations. One obvious difference between both structures is that the denticles of the primary septa extend around the edges of the tubes, forming a three-dimensional conformation (figures 2*g*,*i* and 7*a*,*b*,*i*), instead of the two-dimensional conformation of the crenulations (figure 3*e*,*f*). A three-dimensional conformation introduces changes in the pull-out velocity and direction in different parts of the growing denticles. Another possibility could be differences in the viscosities of the organic membranes in their original state; however, we have no evidence. Complete data on how the septa and their connections with the primary tubes evolve during growth are needed.

Accordingly, the observed dendritic margins are shaped by viscous fingering, which results from the biophysical conditions existing between the intermediate (non-cellular) organic membranes and the living tissue surrounding them. Other dendritic calcified structures in invertebrate molluscs have also been interpreted as the result of similar viscous finger formation processes [38–40].

## 4.4. Final remarks

Shell resistance to crushing in *B. balanus*, which develops interlocking crenulation systems between radii and parietes, was found to be four times higher than in *S. balanoides*, in which the same junctions are smooth [5]. In both *A. psittacus* and *P. perforatus*, there is an interlocking system comparable to that in *B. balanus*, which most likely performs a similar reinforcement function. The teeth of the margin of the alae cannot provide additional strength since they do not form an interlocking system with the relatively smooth surface of the longitudinal abutment. In *N. flosculus*, the high-relief dendrites of both the radius and ala growth margins interlock with the complementary low-relief features of the parietes and sheaths, respectively. Both kinds of growth margins most likely contribute to shell strengthening. The fact that the dendrites are fabricated by a purely physical process implies that the details of the structure are not under strict genetic control (a fabricational feature [41]), i.e. from the functional viewpoint it is not an ideal, but a reasonably efficient design.

Knowing the exact mode of cirriped plate growth, particularly where this takes place, i.e. the margins, is important, not only with regard to the biology of the group, but also for other studies, like biofouling or ocean acidification studies, which use cirripeds as an important study subject.

Data accessibility. Electronic supplementary material, figures S1–S5 and video S1 have been uploaded as part of the electronic supplementary material.
Authors' contributions. A.G.C. conceived and designed the study, acquired and analysed data, wrote the paper. C.S., A.B.R.N. and C.G. acquired and analysed data, revised the paper. N.A.L. conceived and designed the study, acquired and analysed data, revised the paper.
Competing interests. We declare we have no competing interests.

Funding. This research was funded by projects CGL2017-85118-P (A.G.C., C.S. and C.G.) and CGL2015-64683-P (A.B.R.N.) of the Spanish Ministerio de Economía, Industria y Competitividad, the Unidad Científica de Excelencia UCE-PP2016-05 of the University of Granada (A.G.C. and A.B.R.N.) and the Research Group RNM363 of the Junta de Andalucía (A.G.C.). N.A.L. acknowledges support from CONICYT-Chile through grant nos. FONDECYT 1140938, PCI REDES 170106 and PIA ANILLOS ACT172037, for international collaborative research with A.G.C. and A.B.R.N.

Acknowledgements. Luis Sánchez-Tocino (Department of Zoology, University of Granada) provided specimens of *P. perforatus*. Eva M. Enjuto (Andalusian Center of Nanomedicine and Biotechnology, Málaga) carried out methacrylate-embedding and thin-sectioning of specimens of *A. psittacus*.

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
