## [Reviewer comments · Royal Society Open Science]

Review History

RSOS-190458.R0 (Original submission)

Review form: Reviewer 1 (Elizabeth Harper)

Is the manuscript scientifically sound in its present form?

Yes

Are the interpretations and conclusions justified by the results?

Yes

Is the language acceptable?

Yes

Is it clear how to access all supporting data?

Yes

Do you have any ethical concerns with this paper?

No

Have you any concerns about statistical analyses in this paper?

Yes

Recommendation?

Major revision is needed (please make suggestions in comments)

Comments to the Author(s)

This is a very interesting paper. Understanding how barnacles grow with their multiple plates is an intriguing problem – few will have given it much thought but I think that it will be of fairly wide interest to lots of groups of readers once the problems are clear. The methods used are many, sophisticated and extremely appropriate. This is an impressive array of expertise which has been assembled allowing what are rather beautiful results.

I am supportive of publication. My comments are really all about increasing accessibility of the ms. It is a hard read if, like me, you are not familiar with the intricate morphology of barnacles. Accordingly I have three suggestions:

- The reasons for doing this study needs better advertising. There is a missing paragraph on the importance and near ubiquity of barnacles in fouling communities and I am slightly confounded that the very final sentence of the paper is not perhaps right at the beginning!
- Few readers will be familiar with barnacle plate nomenclature. Although the introduction guides us to relevant literature and the supplementary has a useful figure, the main text would be much less foreboding if it had a clear and simple line drawing showing us the names and relationships of the plates. This is not just a cosmetic move. If you don't make it easy for the reader to understand the issues and the results it will be much less easy to get them to read the paper.
- The aims perhaps need setting out more clearly. It is true that there is a series of questions in the introduction – but I think could be set out in a way that explains their relevance to the general problem and provides a framework that the results and discussion can more neatly refer back to.

The findings seem solid – and will be on interest. I am still baffled as to how the organic membrane gets between the plates with no cellular contact but solutions should not have to be found for everything.

I am not quiet sure what it would look like, but again I think that a some kind of summary diagram would help in the final interpretation in the Discussion. The discussion feels rather dense and might benefit (if allowed) from subtitles, perhaps reflecting back to original questions?

Other remarks

What exactly does diametric growth mean?

I would say 'pieces' or 'fragments' rather than 'bits' – in the methods

I am not sure of the use of the term 'carpet' – perhaps line would be better

P16 line 28 – 'transparent' does not seem the right term

P17 – Line 40. This paragraph needs a bit more grounding. It is not clear what 'all cases' are.

Similarly the next para starts 'These traits' which again needs a bit more direction to it.

P19 line 6 'reasonably good design' seems too colloquial

P19 – line 8 – the very final lines! this seems a much better hook for framing the whole study!

Move to earlier in the introduction.

I have not checked the references in detail.

I have no wish to remain anonymous.

Liz Harper

Review form: Reviewer 2 (Andy Gale)

Is the manuscript scientifically sound in its present form?

Yes

Are the interpretations and conclusions justified by the results?

Yes

Is the language acceptable?

Yes

Is it clear how to access all supporting data?

Not Applicable

Do you have any ethical concerns with this paper?

No

Have you any concerns about statistical analyses in this paper?

No

Recommendation?

Accept with minor revision (please list in comments)

Comments to the Author(s)

This is a superbly illustrated, generally well-written paper on balanomorph shell structure which provides a lot of new information on both soft and hard tissues. I hope that another referee can comment on the biomineralisation aspects of the work.

The authors should note that a terminology describing some of the features which they describe and observe in such exquisite detail already exists, and should be referred to. My main concern is that a detailed nomenclature for barnacle microstructure, especially the basis-parietal articulations, already exists in the literature, and the authors do not discuss this or apply it to their material, although they cite a major paper which deals with it (Newman et al. 1967). This terminology should either be applied to the material they describe, or they should explain why they use different terms, and how these relate to those in the literature.

General comment. Gale and Sorensen (2014) provided a back ground to the evolutionary origin of balanomorphs and a new nomenclature for the parietal plates, which are not homologous with those of pedunculate forms. This should be referred to.

Lines 33-35. Note this description only covers the order Thoracica; the Ascothoracica, Rhizocephala, Akentrogonia, Kentrogonida, Cryptophialida and Lithoglyptida and Facetotecta have different, highly diverse life habits, including borers and parasites!

Lines 39-42. Only some balanomorphs have a calcified basis.

Gale, A.S. & Sorensen, A.M. 2014 Origin of the balanomorph barnacles (Cirripedia, Thoracica); new evidence from the Cretaceous of Sweden. *Journal of Systematic Palaeontology*.
<http://dx.doi.org/10.1080/14772019.2014.954824>

Decision letter (RSOS-190458.R0)

24-Jun-2019

Dear Dr Checa,

The editors assigned to your paper ("Articulation and growth of skeletal elements in balanid barnacles (Balanidae, Balanomorphs, Cirripedia)") have now received comments from reviewers. We would like you to revise your paper in accordance with the referee and Associate Editor suggestions which can be found below (not including confidential reports to the Editor). Please note this decision does not guarantee eventual acceptance.

Please submit a copy of your revised paper before 17-Jul-2019. Please note that the revision deadline will expire at 00.00am on this date. If we do not hear from you within this time then it will be assumed that the paper has been withdrawn. In exceptional circumstances, extensions may be possible if agreed with the Editorial Office in advance. We do not allow multiple rounds of revision so we urge you to make every effort to fully address all of the comments at this stage. If deemed necessary by the Editors, your manuscript will be sent back to one or more of the original reviewers for assessment. If the original reviewers are not available, we may invite new reviewers.

- Data accessibility

If you wish to submit your supporting data or code to Dryad (<http://datadryad.org/>), or modify your current submission to dryad, please use the following link:
<http://datadryad.org/submit?journalID=RSOS&manu=RSOS-190458>

- **Competing interests**

- **Authors' contributions**

- **Acknowledgements**

- **Funding statement**

on behalf of Professor Emily Standen (Associate Editor) and Kevin Padian (Subject Editor)
 openscience@royalsociety.org

Subject Editor Comments to Author:

Thanks for your submission. As you can see the reviewers were generally positive, although some issues were raised that we hope you can address specifically. Jargon is a problem but it usually is in anatomical descriptions of all creatures; still, if you can provide a somewhat simplified summary, either in the abstract or conclusion, it will be helpful to non-specialists.

Comments to Author:

Reviewers' Comments to Author:

Reviewer: 1

Comments to the Author(s)

This is a very interesting paper. Understanding how barnacles grow with their multiple plates is an intriguing problem – few will have given it much thought but I think that it will be of fairly wide interest to lots of groups of readers once the problems are clear. The methods used are many, sophisticated and extremely appropriate. This is an impressive array of expertise which has been assembled allowing what are rather beautiful results.

I am supportive of publication. My comments are really all about increasing accessibility of the ms. It is a hard read if, like me, you are not familiar with the intricate morphology of barnacles. Accordingly I have three suggestions:

- The reasons for doing this study needs better advertising. There is a missing paragraph on the importance and near ubiquity of barnacles in fouling communities and I am slightly confounded that the very final sentence of the paper is not perhaps right at the beginning!
- Few readers will be familiar with barnacle plate nomenclature. Although the introduction guides us to relevant literature and the supplementary has a useful figure, the main text would be much less foreboding if it had a clear and simple line drawing showing us the names and relationships of the plates. This is not just a cosmetic move. If you don't make it easy for the reader to understand the issues and the results it will be much less easy to get them to read the paper.
- The aims perhaps need setting out more clearly. It is true that there is a series of questions in the introduction – but I think could be set out in a way that explains their relevance to the general problem and provides a framework that the results and discussion can more neatly refer back to.

The findings seem solid – and will be of interest. I am still baffled as to how the organic membrane gets between the plates with no cellular contact but solutions should not have to be found for everything.

I am not quite sure what it would look like, but again I think that a some kind of summary diagram would help in the final interpretation in the Discussion. The discussion feels rather dense and might benefit (if allowed) from subtitles, perhaps reflecting back to original questions? Other remarks

What exactly does diametric growth mean?

I would say 'pieces' or 'fragments' rather than 'bits' – in the methods

I am not sure of the use of the term 'carpet' – perhaps line would be better

P16 line 28 – 'transparent' does not seem the right term

P17 – Line 40. This paragraph needs a bit more grounding. It is not clear what 'all cases' are. Similarly the next para starts 'These traits' which again needs a bit more direction to it.

P19 line 6 'reasonably good design' seems too colloquial

P19 – line 8 – the very final lines! this seems a much better hook for framing the whole study!
Move to earlier in the introduction.

I have not checked the references in detail.

I have no wish to remain anonymous.

Liz Harper

Reviewer: 2

Comments to the Author(s)

This is a superbly illustrated, generally well-written paper on balanomorph shell structure which provides a lot of new information on both soft and hard tissues. I hope that another referee can comment on the biomineralisation aspects of the work.

The authors should note that a terminology describing some of the features which they describe and observe in such exquisite detail already exists, and should be referred to. My main concern is that a detailed nomenclature for barnacle microstructure, especially the basis-parietal articulations, already exists in the literature, and the authors do not discuss this or apply it to their material, although they cite a major paper which deals with it (Newman et al. 1967). This terminology should either be applied to the material they describe, or they should explain why they use different terms, and how these relate to those in the literature.

General comment. Gale and Sorensen (2014) provided a back ground to the evolutionary origin of balanomorphs and a new nomenclature for the parietal plates, which are not homologous with those of pedunculate forms. This should be referred to.

Lines 33-35. Note this description only covers the order Thoracica; the Ascothoracica, Rhizocephala, Akentrogonia, Kentrogonida, Cryptophialida and Lithoglyptida and Facetotecta have different, highly diverse life habits, including borers and parasites!

Lines 39-42. Only some balanomorphs have a calcified basis.

Gale, A.S. & Sorensen, A.M. 2014 Origin of the balanomorph barnacles (Cirripedia, Thoracica); new evidence from the Cretaceous of Sweden. *Journal of Systematic Palaeontology*.
<http://dx.doi.org/10.1080/14772019.2014>.

Author's Response to Decision Letter for (RSOS-190458.R0)

See Appendix A.

RSOS-190458.R1 (Revision)

Review form: Reviewer 1 (Elizabeth Harper)

Is the manuscript scientifically sound in its present form?

Yes

Are the interpretations and conclusions justified by the results?

Yes

Is the language acceptable?

Yes

Do you have any ethical concerns with this paper?

No

Recommendation?

Accept with minor revision (please list in comments)

Comments to the Author(s)

p2 - line 58 - there is something vaguely curious about the wording in the sentence which begins 'When measuring success with the...' - although it is not totally clear why. May be it should be 'by' and not 'with' or perhaps it should be 'criteria'

p3, line 6 - really 'the' most important? Perhaps say 'a' or 'one of the '

p13, line 47 - 'vary' might be better than 'change' as that suggests something ongoing?

p13, line 58 - fiber spelling inconsistent with rest of the ms

Decision letter (RSOS-190458.R1)

02-Aug-2019

Dear Dr Checa:

On behalf of the Editors, I am pleased to inform you that your Manuscript RSOS-190458.R1 entitled "Articulation and growth of skeletal elements in balanid barnacles (Balanidae, Balanomorpha, Cirripedia)" has been accepted for publication in Royal Society Open Science subject to minor revision in accordance with the referee suggestions. Please find the referees' comments at the end of this email.

The reviewers and Subject Editor have recommended publication, but also suggest some minor revisions to your manuscript. Therefore, I invite you to respond to the comments and revise your manuscript.

- Ethics statement

- Data accessibility

<http://datadryad.org/submit?journalID=RSOS&manu=RSOS-190458.R1>

- Competing interests

- Authors' contributions

- Acknowledgements

- Funding statement

Because the schedule for publication is very tight, it is a condition of publication that you submit the revised version of your manuscript before 11-Aug-2019. Please note that the revision deadline will expire at 00.00am on this date. If you do not think you will be able to meet this date please let me know immediately.

on behalf of Professor Emily Standen (Associate Editor) and Kevin Padian (Subject Editor)
openscience@royalsociety.org

Associate Editor Comments to Author (Professor Emily Standen):

Dear Dr. Checa,

We are happy to see that there are only a few very minor changes that need to be addressed in this latest review of your manuscript entitled 'Articulation and growth of skeletal elements in balanid barnacles'. If you could please address these comments and return your manuscript to us we would be grateful.

Sincerely,
EMS

Reviewer comments to Author:

Reviewer: 1

p2 - line 58 - there is something vaguely curious about the wording in the sentence which begins 'When measuring success with the...' - although it is not totally clear why. May be it should be 'by' and not 'with' or perhaps it should be 'criteria'

p3, line 6 - really 'the' most important? Perhaps say 'a' or 'one of the '

p13, line 47 - 'vary' might be better than 'change' as that suggests something ongoing?

p13, line 58 - fiber spelling inconsistent with rest of the ms

Author's Response to Decision Letter for (RSOS-190458.R1)

See Appendix B.

Decision letter (RSOS-190458.R2)

08-Aug-2019

Dear Dr Checa,

I am pleased to inform you that your manuscript entitled "Articulation and growth of skeletal elements in balanid barnacles (Balanidae, Balanomorpha, Cirripedia)" is now accepted for publication in Royal Society Open Science.

on behalf of Professor Emily Standen (Associate Editor) and Kevin Padian (Subject Editor)
openscience@royalsociety.org

Appendix A

First of all, we want to express our gratitude to the reviewers for their positive and useful comments. We have tried to incorporate all of them in the present version and hope they will be pleased with the solutions we have adopted.

All changes have been highlighted in red in the 'with changes highlighted' version

Reviewer: 1

This is a very interesting paper. Understanding how barnacles grow with their multiple plates is an intriguing problem – few will have given it much thought but I think that it will be of fairly wide interest to lots of groups of readers once the problems are clear. The methods used are many, sophisticated and extremely appropriate. This is an impressive array of expertise which has been assembled allowing what are rather beautiful results.

I am supportive of publication. My comments are really all about increasing accessibility of the ms. It is a hard read if, like me, you are not familiar with the intricate morphology of barnacles. Accordingly I have three suggestions:

- The reasons for doing this study needs better advertising. There is a missing paragraph on the importance and near ubiquity of barnacles in fouling communities and I am slightly confounded that the very final sentence of the paper is not perhaps right at the beginning!

Following this suggestion, we have introduced a paragraph (2nd paragraph of Introduction, pages 2-3) about the importance of acorn barnacles in littoral communities, their importance as biofouling agents and their susceptibility to climatic change. There are two sentences at the end of the Introduction (beginning of page 5), which also allude to the incidence of our study on these general aspects.

- Few readers will be familiar with barnacle plate nomenclature. Although the introduction guides us to relevant literature and the supplementary has a useful figure, the main text would be much less foreboding if it had a clear and simple line drawing showing us the names and relationships of the plates. This is not just a cosmetic move. If you don't make it easy for the reader to understand the issues and the results it will be much less easy to get them to read the paper.

In principle, Figure 1 was initially intended to cover this aspect. The main elements of balanids are been labelled, and the information provided is extensive. Nevertheless, the images of Figure 1a have been changed to make it even more illustrative.

- The aims perhaps need setting out more clearly. It is true that there is a series of questions in the introduction – but I think could be set out in a way that explains their relevance to the general problem and provides a framework that the results and discussion can more neatly refer back to.

Right, a final statement has been introduced at the end of the Discussion (page 20), which refers to the additions made in the Introduction (see above).

The findings seem solid – and will be on interest. I am still baffled as to how the organic membrane gets between the plates with no cellular contact but solutions should not have to be found for everything.

I am not quite sure what it would look like, but again I think that a some kind of summary diagram would help in the final interpretation in the Discussion. The discussion feels rather

dense and might benefit (if allowed) from subtitles, perhaps reflecting back to original questions?

It is true that the Discussion seems a bit hard to go through. To make the information more accessible, we have introduced a series of subsections (4.1. to 4.4.), which separate the main aspects.

Other remarks

- What exactly does diametric growth mean?

In this kind of growth, there is increase in the periphery of both the basis and apex, thus leading the self-similarity. In monometric growth, only the basal periphery increases, with the consequence that the aperture becomes proportionately reduced. This is now better specified (page 3, end of second paragraph).

- I would say 'pieces' or 'fragments' rather than 'bits' –

Changed to 'Pieces' (page 6, beginning of subsection 2.4.)

- in the methods I am not sure of the use of the term 'carpet' – perhaps line would be better.

Changed to 'lined' (page 13, beginning of 2nd paragraph).

- P16 line 28 – 'transparent' does not seem the right term

'transparent' has been removed (page 17, end of 1st paragraph).

- P17 – Line 40. This paragraph needs a bit more grounding. It is not clear what 'all cases' are. Similarly the next para starts 'These traits' which again needs a bit more direction to it.

This is now indicated as 'the lateral and basal boundaries between plates' (page 17, beginning of subsection 4.3.), and 'These dendritic traits', referred to immediately above (page 18, 1st line).

- P19 line 6 'reasonably good design' seems too colloquial

Changed to 'reasonably efficient design'.

- P19 – line 8 – the very final lines! this seems a much better hook for framing the whole study! Move to earlier in the introduction.

This is now moved and more widely elaborated in the Introduction (see above), although a small reference to the statements made in the Introduction remains here.

I have not checked the references in detail.

I have no wish to remain anonymous.

Liz Harper

Reviewer: 2

This is a superbly illustrated, generally well-written paper on balanomorph shell structure which provides a lot of new information on both soft and hard tissues. I hope that another referee can comment on the biomineralisation aspects of the work.

The authors should note that a terminology describing some of the features which they describe and observe in such exquisite detail already exists, and should be referred to. My main concern is that a detailed nomenclature for barnacle microstructure, especially the basis-parietal articulations, already exists in the literature, and the authors do not discuss this or apply it to their material, although they cite a major paper which deals with it (Newman et al. 1967). This terminology should either be applied to the material they describe, or they should explain why they use different terms, and how these relate to those in the literature.

Right, the references from which we have adopted the terminology are now explicitly cited (our references [1], [15], [16], [17] and [18]; page 3, beginning of 3rd paragraph).

General comment. Gale and Sorensen (2014) provided a back ground to the evolutionary origin of balanomorphs and a new nomenclature for the parietal plates, which are not homologous with those of pedunculate forms. This should be referred to.

Although we knew the reference, we had not paid enough attention to it. The new terms introduced by Gale and Sørensen (2014) (our new reference [18]) are discussed (page 3, 3rd to 5th sentences of 3rd paragraph) and adopted throughout (figures have been re-labelled accordingly).

- Lines 33-35. Note this description only covers the order Thoracica; the Ascothoracica, Rhizocephala, Akentrogonia, Kentrogonida, Cryptophialida and Lithoglyptida and Facetotecta have different, highly diverse life habits, including borers and parasites!

Right, this is now indicated in the first two sentences of the Introduction (page 2).

- Lines 39-42. Only some balanomorphs have a calcified basis.

We now state that this is the case for balanids (page 4, last sentence of 1st paragraph).

Gale, A.S. & Sorensen, A.M. 2014 Origin of the balanomorph barnacles (Cirripedia, Thoracica); new evidence from the Cretaceous of Sweden. Journal of Systematic Palaeontology. <http://dx.doi.org/10.1080/14772019.2014.954824>

Additional comment

The previous subsection on microstructures (formerly 4.2.) has now been expanded to introduce the recent observation that the microstructure is different depending on whether the growing margin is in contact with living tissue or with acellular membranes. The corresponding figure (formerly figure 3) has been changed to illustrate better this fact. After this, both the subsection and figure needed a new placement. The subsection is now subsection 3.4. (pages 13-14), and the figure is now figure 8. This aspect is also alluded to in the Discussion (page 17, 2nd paragraph).

Appendix B

First of all, we want to express our gratitude to the reviewers for their positive and useful comments.

Reviewer: 1

p2 - line 58 - there is something vaguely curious about the wording in the sentence which begins 'When measuring success with the...' - although it is not totally clear why. May be it should be 'by' and not 'with' or perhaps it should be 'criteria'

Right! Changed to "by the criteria"

p3, line 6 - really 'the' most important'? Perhaps say 'a' or 'one of the '

Changed accordingly: "one of the most successful"

p13, line 47 - 'vary' might be better than 'change' as that suggests something ongoing?

Changed to "vary"

p13, line 58 - fiber spelling inconsistent with rest of the ms.

Changed to "fibre". We have checked that this was the only instance.